# Off-Policy Evaluation for Human Feedback

Qitong Gao*  Ge Gao†  Juncheng Dong*  Vahid Tarokh*  Min Chi†  Miroslav Pajic*

## Abstract

Off-policy evaluation (OPE) is important for closing the gap between offline training and evaluation of reinforcement learning (RL), by estimating performance and/or rank of target (evaluation) policies using offline trajectories only. It can improve the safety and efficiency of data collection and policy testing procedures in situations where online deployments are expensive, such as healthcare. However, existing OPE methods fall short in estimating human feedback (HF) signals, as HF may be conditioned over multiple underlying factors and is only sparsely available; as opposed to the agent-defined environmental rewards (used in policy optimization), which are usually determined over parametric functions or distributions. Consequently, the nature of HF signals makes extrapolating accurate OPE estimations to be challenging. To resolve this, we introduce an OPE for HF (OPEHF) framework that revives existing OPE methods in order to accurately evaluate the HF signals. Specifically, we develop an immediate human reward (IHR) reconstruction approach, regularized by environmental knowledge distilled in a latent space that captures the underlying dynamics of state transitions as well as issuing HF signals. Our approach has been tested over *two real-world experiments*, adaptive *in-vivo* neurostimulation and intelligent tutoring, as well as in a simulation environment (visual Q&A). Results show that our approach significantly improves the performance toward estimating HF signals accurately, compared to directly applying (variants of) existing OPE methods.

## 1 Introduction

Off-policy evaluation (OPE) aims to estimate the performance of reinforcement learning (RL) policies using only a fixed set of offline trajectories [61], *i.e.*, without online deployments. It is considered to be a critical step in closing the gap between offline RL training and evaluation, for environments and systems where online data collection is expensive or unsafe. Specifically, OPE facilitates not only offline evaluation of the safety and efficacy of the policies ahead of online deployment, but policy selection as well; this allows one to maximize the efficiency when online data collection is possible, by identifying and deploying the policies that are more likely to result in higher returns. OPE has been used in various application domains including healthcare [68, 53, 23, 22], robotics [15, 18, 24], intelligent tutoring [64, 45, 17], recommendation systems [50, 43].

The majority of existing OPE methods focus on evaluating the policies' performance defined over the *environmental* reward functions which are mainly designed for use in policy optimization (training). However, as an increasing number of offline RL frameworks are developed for human-involved systems [64, 45, 1, 48, 16], existing OPE methods lack the ability to estimate how human users would evaluate the policies, *e.g.*, ratings provided by patients (on a scale of 1-10) over the procedure facilitated by automated surgical robots; as human feedback (HF) can be noisy and conditioned over various confounders that could be difficult to be captured explicitly [53, 7, 44]. For example, patient satisfaction over a specific diabetes therapy may vary across the cohort, depending on many subjective

---

*Duke University. Durham, NC, USA. Contact: {qitong.gao, miroslav.pajic}@duke.edu.
†North Carolina State University. Raleigh, NC, USA.

37th Conference on Neural Information Processing Systems (NeurIPS 2023).

factors, such as personal preferences and activity level of the day, while participating in the therapy, in addition to the physiological signals (*e.g.*, blood sugar level, body weight) that are more commonly used as the sources for determining environmental rewards toward policy optimization [70, 33, 21, 19]. Moreover, the environmental rewards are sometimes discrete to ensure optimality of the learned policies [67], which further reduces its correlation against HF signals.

In this work, we introduce the OPE for human feedback (OPEHF) framework that revives existing OPE approaches in the context of evaluating HF from offline data. Specifically, we consider the challenging scenario where the HF signal is only provided at the end of each episode – *i.e.*, no per-step HF signals, referred to as *immediate human rewards* (IHRs) below, are provided – benchmarking the common real-world situations where the participants are allowed to rate the procedures only at the end of the study. The goal is set to estimate the end-of-episode HF signals, also referred to as *human returns*, over the target (evaluation) policies, using a fixed set of offline trajectories collected over some behavioral policies. To facilitate OPEHF, we introduce an approach that first maps the human return back to the sequence of IHRs, over the horizon, for each trajectory. Specifically, this follows from optimizing over an objective that consists of a necessary condition where the cumulative discounted sum of IHRs should equal the human return, as well as a regularization term that limits the discrepancy of the reconstructed IHRs over state-action pairs that are determined similar over a latent representation space into which environmental transitions and rewards are encoded. At last, this allows for the use of any existing OPE methods to process the offline trajectories with reconstructed IHRs and estimate human returns under target policies.

Our main contributions are tri-fold. (***i***) We introduce a novel OPEHF framework that revives existing OPE methods toward accurately estimating highly sparse HF signals (provided only at the end of each episode) from offline trajectories, through IHRs reconstruction. (***ii***) Our approach does not require the environmental rewards and the HF signals to be strongly correlated, benefiting from the design where both signals are encoded to a latent space regularizing the objective for reconstructions of IHRs, which is justified empirically over real-world experiments. (***iii***) Two *real-world experiments*, *i.e.*, adaptive *in-vivo* neurostimulation for the treatment of Parkinson's disease and intelligent tutoring for computer science students in colleges, as well as one simulation environment (*i.e.*, visual Q&A), facilitated the thorough evaluation of our approach; various degrees of correlations between the environment rewards and HF signals existed across the environments, as well as the varied coverage of the state-action space provided by offline data over sub-optimal behavioral policies, imposing different levels of challenges for OPEHF.

## 2 Off-Policy Evaluation for Human Feedback (OPEHF)

In this section, we introduce an OPEHF framework that allows for the use of existing OPE methods to estimate the *human returns* that are available only at the end of each episode, with IHRs remaining unknown. This is in contrast to the goal of classic OPE that only estimates the *environmental* returns following the user-defined reward function used in the policy optimization phase. A brief overview of existing OPE methods can be found in Appendix C.

### 2.1 Problem Formulation

We first formulate the human-involved MDP (HMDP), which is a tuple $\mathcal{M} = (\mathcal{S}, \mathcal{A}, \mathcal{P}, R, R^{\mathcal{H}}, s_0, \gamma)$, where $\mathcal{S}$ is the set of states, $\mathcal{A}$ the set of actions, $\mathcal{P} : \mathcal{S} \times \mathcal{A} \to \mathcal{S}$ is the transition distribution usually captured by probabilities $p(s_t|s_{t-1}, a_{t-1})$, $R : \mathcal{S} \times \mathcal{A} \to \mathbb{R}$ is the *environmental* reward function, $R^{\mathcal{H}}(r^{\mathcal{H}}|s, a)$ is the *human* reward distribution from which the IHR $r_t^{\mathcal{H}} \sim R^{\mathcal{H}}(\cdot|s_t, a_t)$ are sampled, $s_0$ is the initial state sampled from the initial state distribution $p(s_0)$, and $\gamma \in [0, 1)$ is the discounting factor. Note that we set the IHRs to be determined probabilistically, as opposed to the environmental rewards $r_t = R(s_t, a_t)$ that are deterministic; this is due to the fact that many underlying factors may affect the feedback provided by humans [53, 7, 44], as we have also observed while performing human-involved experiments (see Appendix D). Finally, the agent interacts with the MDP following some policy $\pi(a|s)$ that defines the probabilities of taking action $a$ at state $s$.

In this work, we make the following assumption over $R$ and $R^{\mathcal{H}}$.

**Assumption 1** (Unknown IHRs)**.** *We assume that the immediate environmental reward function $R$ is known and $R(s, a)$ can be obtained for any state-action pairs in $\mathcal{S} \times \mathcal{A}$. Moreover, the IHR distribution $R^{\mathcal{H}}$ is assumed to be unknown, i.e., $r^{\mathcal{H}} \sim R^{\mathcal{H}}(\cdot|s, a)$ are unobservable, for all $(s, a) \in \mathcal{S} \times \mathcal{A}$.*

*Instead, the cumulative human return $G_{0:T}^{\mathcal{H}}$, defined over $R^{\mathcal{H}}$, is given at the end of each trajectory, i.e., $G_{0:T}^{\mathcal{H}} = \sum_{t=0}^{T} \gamma^t r_t^{\mathcal{H}}$, with $T$ being the horizon and $r_t^{\mathcal{H}} \sim R^{\mathcal{H}}(\cdot|s_t, a_t)$.*

The assumption above follows the fact that human feedback (HF) is not available until the end of each episode, as opposed to immediate rewards that can be defined over the environment and evaluated for any $(s_t, a_t)$ pairs at any time. This is especially true in environments such as healthcare where the clinical treatment outcome is not foreseeable until a therapeutic cycle is completed, or in intelligent tutoring where the overall gain from students over a semester is mostly reflected by the final grades. Note that although the setup can be generalized to the scenario where HF can be sparsely obtained over the horizon, we believe that issuing the HF only at the end of each trajectory leads to a more challenging setup for OPE. Consequently, the goal of OPEHF can be formulated as follows.

**Problem 1** (Objective of OPEHF). *Given offline trajectories collected by some behavioral policy $\beta$, $\rho^{\beta} = \{\tau^{(0)}, \tau^{(1)}, \ldots, \tau^{(N-1)}| \ a_t \sim \beta(a_t|s_t)\}$, with $\tau^{(i)} = [(s_0^{(i)}, a_0^{(i)}, r_0^{(i)}, r_0^{\mathcal{H}(i)}, s_1^{(i)}), \ldots, (s_{T-1}^{(i)}, a_{T-1}^{(i)}, r_{T-1}^{(i)}, r_{T-1}^{\mathcal{H}(i)}, s_T^{(i)}), G_{0:T}^{\mathcal{H}(i)}]$ being a single trajectory, $N$ the total number of offline trajectories, and $r_t^{\mathcal{H}}$'s being* **unknown***, the objective is to estimate the expected total human return over the unknown state-action visitation distribution $\rho^{\pi}$ of the target (evaluation) policy $\pi$, i.e., $\mathbb{E}_{(s,a)\sim\rho^{\pi}, r^{\mathcal{H}}\sim R^{\mathcal{H}}}\left[\sum_{t=0}^{T} \gamma^t r_t^{\mathcal{H}}\right]$.*

### 2.2 Reconstrunction of IHRs for OPEHF

We emphasize that the human returns are only issued at the end of each episode, with IHRs remaining unknown. One can set all IHRs from $t=0$ to $t=T-2$ to be zeros (*i.e.*, $r_{0:T-2}^{\mathcal{H}} = 0$), and *rescale* the cumulative human return to be the IHR at the last step (*i.e.*, $r_{T-1}^{\mathcal{H}} = G_{0:T}^{\mathcal{H}}/\gamma^{T-1}$), to allow the use of existing OPE methods toward OPEHF (Problem 1). However, the sparsity over $r^{\mathcal{H}}$'s here may impose difficulties for OPE to estimate the human returns accurately over the target policies. For OPEHF, we start by showing that for the per-decision importance sampling (PDIS) method – a variance-reduction variant of the importance sample (IS) family of OPE methods [61] – if IHRs *were to be available*, they could reduce the variance in the estimation compared to the rescale approach above.

Recall that the PDIS estimator follows $\hat{G}_{PDIS}^{\pi} = \frac{1}{N}\sum_{i=1}^{N-1}\sum_{t=0}^{T-1} \gamma^t \omega_{0:t}^{(i)} r_t^{\mathcal{H}()}$, where $\omega_{0:t}^{(i)} = \prod_{k=0}^{t} \frac{\pi(a_k^{(i)}|s_k^{(i)})}{\beta(a_k^{(i)}|s_k^{(i)})}$ are the PDIS weights for offline trajectory $\tau^{(i)}$. Moreover, the estimator of the rescale approach[3] above is $\hat{G}_{Rescale}^{\pi} = \frac{1}{N}\sum_{i=1}^{N-1} \omega_{0:T-1}^{(i)} G_{0:T}^{\mathcal{H}(i)}$, which is equivalent to the vanilla IS estimator [61, 72]. We now show the variance reduction property of $\hat{G}_{PDIS}^{\pi}$ in the context of OPEHF.

**Proposition 1.** *Assume that (i) $\mathbb{E}[r_t^{\mathcal{H}}] \geq 0$, and (ii) given the horizon $T$, consider any $1 \leq t+1 \leq k \leq T$ of any offline trajectory $\tau$, $\omega_{0:k}$ and $r_t^{\mathcal{H}}\omega_{0:k}$ are positively correlated. Then, $\mathbb{V}(\hat{G}_{PDIS}^{\pi}) \leq \mathbb{V}(\hat{G}_{Rescale}^{\pi})$, with $\mathbb{V}(\cdot)$ representing the variance.*

The proof can be found in Appendix A. Assumption (*i*) can be easily satisfied in the real world, as HF signals are usually quantified as positive values, *e.g.*, ratings (1-10) provided by participants. Assumption (*ii*) is most likely to be satisfied when the target policies do not visit low-return regions substantially [46], which is a pre-requisite for testing RL policies in human-involved environments as initial screening are usually required to filter the ones that could potentially pose risks to participants [57].

Besides IS, doubly robust (DR) [71, 34, 69, 12] and fitted Q-evaluation (FQE) [40] methods require learning value functions. Sparsity of rewards (following the rescale approach above) in the offline dataset may lead to poorly learned value functions [74], considering that the offline data in OPE is usually fixed (*i.e.*, no new samples can be added), and are often generated by behavioral policies that are sub-optimal, which results in limited coverage of the state-action space. Limited availabilities of environment-policy interactions (*e.g.*, clinical trials) further reduce the scale of the exploration and therefore limit the information that can be leveraged toward obtaining accurate value function approximations.

**Reconstruction of IHRs.** To address this challenge, our approach aims to project the end-of-episode human returns back to each environmental step, *i.e.*, to learn a mapping $f_\theta(\tau, G_{0:T}^{\mathcal{H}}) : (\mathcal{S} \times \mathcal{A})^T \times$

---

[3]We call it the *rescale approach* instead of vanilla IS as the idea behind also generalizes to non-IS methods.

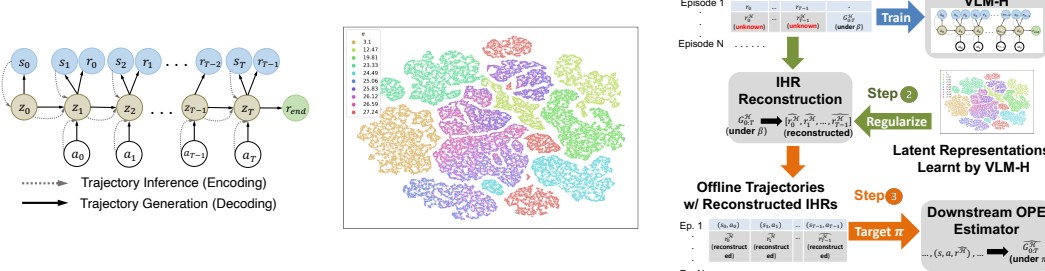

Figure 1: (**Left**) Architecture of the variational latent model with human returns (VLM-H). (**Mid**) Illustration of the clustering behavior in the latent space using $t$-SNE visualization [73], where the encoded state-action pairs (output by the encoder of VLM-H) are in general clustered together if they are generated by policies with similar human returns (shown in the legend at the top left). (**Right**) Diagram summarizing the pipeline of the OPEHF framework.

$\mathbb{R} \to \mathbb{R}^T$, parameterized by $\theta$, that maximizes the sum of log-likelihood of the estimated IHRs, $[\hat{r}_0^{\mathcal{H}}, \ldots, \hat{r}_{T-1}^{\mathcal{H}}]^\mathsf{T} \sim f_\theta(\tau, G_{0:T}^{\mathcal{H}})$, following $\max_\theta \frac{1}{N} \sum_{i=0}^{N-1} \sum_{t=0}^{T-1} \log p(\hat{r}_t^{\mathcal{H}} = r_t^{\mathcal{H}(i)} | \theta, \tau^{(i)}, G_{0:T}^{\mathcal{H}(i)})$, where $G_{0:T}^{\mathcal{H}(i)}$ and $r_t^{\mathcal{H}(i)}$'s are respectivelly the human return and IHRs (unknown) of the $i$-th trajectory in the offline dataset $\rho^\beta$, and $N$ is the total number of trajectories in $\rho^\beta$. Given that the objective above is intractable due to unknown $r_t^{\mathcal{H}(i)}$'s, we introduce a surrogate objective

$$\max_\theta \frac{1}{N} \sum_{i=0}^{N-1} \left[ \log p\left( \sum_{t=0}^{T-1} \gamma^t \hat{r}_t^{\mathcal{H}} = G_{0:T}^{\mathcal{H}(i)} | \theta, \tau^{(i)}, G_{0:T}^{\mathcal{H}(i)} \right) - C \cdot \mathcal{L}_{regu}(\hat{r}_{0:T-1}^{\mathcal{H}} | \theta, \tau^{(i)}, G_{0:T}^{\mathcal{H}(i)}) \right]. \quad (1)$$

Here, the *first term* is a necessary condition for $\hat{r}_t^{\mathcal{H}}$'s to be valid for estimating $r_t^{\mathcal{H}}$'s, as they should sum to $G_{0:T}^{\mathcal{H}}$. Since many solutions may exist if one only optimizes over the first term, the *second term* $\mathcal{L}_{regu}$ serves as a regularization that imposes constraints on $r_t^{\mathcal{H}}$'s to follow the properties specific to their corresponding state-action pairs; *e.g.*, $(s, a)$ pairs that are similar to each other in a representation space, defined over the state-action visitation space, tend to yield similar immediate rewards [18].

The detailed regularization technique is introduced in sub-section below. Practically, we choose $f_\theta$ to be a bi-directional long-short term memory (LSTM) [32], since the reconstruction of IHRs can leverage information from both previous and subsequent steps as provided in the offline trajectories.

## 2.3  Reconstruction of IHRs over Latent Representations (RILR) for OPEHF

Now, we introduce the regularization technique for the reconstruction of IHRs, *i.e.*, reconstructing IHRs over latent representations (RILR). Specifically, we leverage the representations captured by variational auto-encoders (VAEs) [35], learned over $\hat{\rho}^\beta$, to regularize the reconstructed IHRs, $\hat{r}_t^{\mathcal{H}}$.

VAEs have been adapted toward learning a compact latent space over offline state-action visitations, facilitating both offline policy optimization [42, 81, 65, 27, 26, 28] and OPE [18]. In this work, we specifically consider building on the variational latent model (VLM) proposed in [18] since it is originally proposed to facilitate OPE, as opposed to others that mainly use knowledge captured in the latent space to improve sample efficiency for policy optimization. Moreover, the VLM has shown to be effective for learning an expressive representation space, where the encoded state-action pairs are clustered well in the latent space, as measured by the difference over the returns of the policies from which the state-action pairs are sampled; see Figure 1 (mid) which uses $t$-SNE to visualize the encoded state-action pairs in trajectories collected from a visual Q&A environment (Appendix E).

Note that VLM originally does not account for HF signals (neither $r_t^{\mathcal{H}}$'s nor $G_{0:T}^{\mathcal{H}}$'s), so we introduce the variational latent model with human returns (VLM-H) below, building on the architecture introduced in [18]. VLM-H consists of a prior $p(z)$ over the latent variables $z \in \mathcal{Z} \subset \mathbb{R}^L$, with $\mathcal{Z}$ representing the latent space and $L$ the dimension, along with a variational encoder $q_\psi(z_t | z_{t-1}, a_{t-1}, s_t)$, a decoder $p_\phi(z_t, s_t, r_{t-1} | z_{t-1}, a_{t-1})$ for generating per-step transitions (over both state-action and latent space), and a separate decoder $p_\phi(G_{0:T}^{\mathcal{H}} | z_T)$ for the reconstruction of the human returns at the end of each episode. Note that encoders and decoders are parameterized by $\psi$ and $\phi$ respectively. The overall architecture is illustrated in Figure 1 (left).

**Trajectory inference (encoding).** VLM-H's encoder approximates the intractable posterior $p(z_t|z_{t-1}, a_{t-1}, s_t) = \frac{p(z_{t-1}, a_{t-1}, z_t, s_t)}{\int_{z_t \in \mathcal{Z}} p(z_{t-1}, a_{t-1}, z_t, s_t) dz_t}$, by avoiding to integrate over the unknown latent space *a priori*. The inference (or encoding) process can be decomposed as, *i.e.*, $q_\psi(z_{0:T}|s_{0:T}, a_{0:T-1}) = q_\psi(z_0|s_0) \prod_{t=1}^{T} q_\psi(z_t|z_{t-1}, a_{t-1}, s_t)$; here, $q_\psi(z_0|s_0)$ encodes initial states $s_0$ into latent variables $z_0$, and $q_\psi(z_t|z_{t-1}, a_{t-1}, s_t)$ captures all subsequent environmental transitions in the latent space over $z_t$'s. In general, both $q_\psi$'s are represented as diagonal Gaussian distributions[4] with mean and variance determined by neural network $\psi$, as in [18, 42, 27, 26, 28].

**Trajectory generation (decoding).** The generative (or decoding) process follows, *i.e.*, $p_\phi(z_{1:T}, s_{0:T}, r_{0:T-1}, G_{0:T}^{\mathcal{H}}|z_0, \pi) = p_\phi(G_{0:T}^{\mathcal{H}}|z_T) \cdot \prod_{t=1}^{T} p_\phi(z_t|z_{t-1}, a_{t-1})p_\phi(r_{t-1}|z_t) \cdot \prod_{t=0}^{T} p_\phi(s_t|z_t)$; here, $p_\phi(z_t|z_{t-1}, a_{t-1})$ enforces the transition of latent variables $z_t$ over time, $p_\phi(s_t|z_t)$ and $p_\phi(r_{t-1}|z_t)$ are used to sample the states and immediate *environmental* rewards, while $p_\phi(G_{0:T}^{\mathcal{H}}|z_T)$ generates the *human return* issued at the end of each episode. Note that here we still use the VLM-H to capture environmental rewards, allowing the VLM-H to formulate a latent space that captures as much information about the dynamics underlying the environment as possible. All $p_\phi$'s are represented as diagonal Gaussians[5] with parameters determined by network $\phi$.

To train $\phi$ and $\psi$, one can maximize the evidence lower bound (ELBO) of the joint log-likelihood $\log p_\phi(s_{0:T}, r_{0:T-1}, G_{0:T}^{\mathcal{H}}|\phi, \psi, \rho^\beta)$, *i.e.*,

$$\max_{\psi,\phi} \quad \mathbb{E}_{q_\psi} \Big[ \log p_\phi(G_{0:T}^{\mathcal{H}}|z_T) + \sum_{t=0}^{T} \log p_\phi(s_t|z_t) + \sum_{t=1}^{T} \log p_\phi(r_{t-1}|z_t)$$

$$- KL\big(q_\psi(z_0|s_0)||p(z_0)\big) - \sum_{t=1}^{T} KL\big(q_\psi(z_t|z_{t-1}, a_{t-1}, s_t)||p_\phi(z_t|z_{t-1}, a_{t-1})\big)\Big]; \quad (2)$$

the first three terms are the log-likelihoods of reconstructing the human return, states, and environmental rewards, and the two terms that follow are Kullback-Leibler (KL) divergence [38] regularizing the inferred posterior $q_\psi$. Derivation of the ELBO can be found in Appendix B. In practice, if $\phi$ and $\psi$ are chosen to be recurrent networks, one can also regularize the hidden states of $\phi, \psi$ by including the additional regularization term introduced in [18].

**Regularizing the reconstruction of IHRs.** Existing works have shown that the latent space not only facilitates the generation of synthetic trajectories but demonstrated that the latent encodings of state-action pairs form clusters, over some measures in the latent space [73], if they are rolled out from policies that lead to similar returns [42, 18]. As a result, we regularize $\hat{r}_t^{\mathcal{H}}$ following

$$\min_\theta \mathcal{L}_{regu}(\hat{r}_t^{\mathcal{H}}|\theta, \psi, s_{0:t}^{(i)}, a_{0:t-1}^{(i)}, G_{0:T}^{\mathcal{H}(i)}) = \sum_{j \in \mathcal{J}} -\log p(\hat{r}_t^{\mathcal{H}} = (1-\gamma)G_{0:T}^{\mathcal{H}(j)}|\theta, \psi, s_{0:t'}^{(j)}, a_{0:t'-1}^{(j)}, G_{0:T}^{\mathcal{H}(j)})$$

$$(3)$$

for each step $t$; here, $(s_{0:t}^{(i)}, a_{0:t-1}^{(i)}) \in \tau^{(i)} \sim \rho^\beta$, $\mathcal{J} = \{j_0, \ldots, j_{K-1}\}$ are the indices of offline trajectories that correspond to the latent encodings $\{z_{t'}^{(j_k)} \sim q_\psi(\cdot|s_{0:t'}^{(j_k)}, a_{0:t'-1}^{(j_k)})|j_k \in \mathcal{J}, t' \in [0, T-1]\}$ that are $K$-neighbours of the latent encoding $z_t^{(i)}$ pertaining to $(s_{0:t}^{(i)}, a_{0:t-1}^{(i)})$, defined over some similarity/distance function $d(\cdot||\cdot)$, following, *i.e.*,

$$\min_{j_k \in \mathcal{J}} \sum_{k=0}^{K-1} d(z_t^{(i)}||z_{t'}^{(j_k)}), \quad \text{s.t. } z_{t'}^{(j_k)} \text{'s corresponding } \big(s_{0:t'}^{(j_k)}, a_{0:t'-1}^{(j_k)}\big) \in \tau^{(j_k)} \sim \rho^\beta. \quad (4)$$

In practice, we choose $d(\cdot||\cdot)$ to follow stochastic neighbor embedding (SNE) similarities [73], as it has been shown effective for capturing Euclidean distances in high-dimensional space [75].

**Overall objective of RILR for OPEHF.** As a result, by following (1) and leveraging the $\mathcal{L}_{regu}$ from (3) above, the objective for reconstructing the IHRs is set to be, *i.e.*,

$$\max_\theta \frac{1}{N} \sum_{i=0}^{N-1} \Big[ \log p\Big(\sum_{t=0}^{T-1} \gamma^t \hat{r}_t^{\mathcal{H}} = G_{0:T}^{\mathcal{H}(i)}|\theta, \tau^{(i)}, G_{0:T}^{\mathcal{H}(i)}\Big) - C \cdot \sum_{t=0}^{T-1} \mathcal{L}_{regu}(\hat{r}_t^{\mathcal{H}}|\theta, \psi, s_{0:t}^{(i)}, a_{0:t-1}^{(i)}, G_{0:T}^{\mathcal{H}(i)})\Big].$$

$$(5)$$

---

[4]This helps facilitate an orthogonal basis of the latent space, which would improve the expressiveness of the model.

[5]If needed, one can project the states over to the orthogonal basis, to ensure that they follow a diagonal co-variance.

**Move from RILR to OPEHF.** In what follows, one can leverage any existing OPE methods to take as inputs the offline trajectories, with the immediate environmental rewards $r_t$'s replaced by the reconstructed IHRs $\hat{r}_t^{\mathcal{H}}$'s, to achieve the OPEHF's objective (Problem 1). Moreover, our method does not require the IHRs to be correlated with the environmental rewards, as the VLM-H learns to reconstruct both by sampling from two independent distributions, $p_\phi(r_{t-1}|z_t)$ and $p_\phi(G_{0:T}^{\mathcal{H}}|z_T)$ respectively, following (2); this is also illustrated empirically over the experiments introduced below (Sections 3), where exceedingly low correlations are found in specific scenarios.

The overall pipeline summarizing our method is shown in Figure 1 (right).

## 3 Real-World Experiments with Human Participants

In this section, we validate the OPEHF framework introduced above over two real-world experiments, adaptive neurostimulation, and intelligent tutoring. Specifically, we consider four types of OPE methods to be used as the downstream estimator following the RILR step (Section 2.3), including per-decision importance sampling (IS) with behavioral policy estimation [30], doubly robust (DR) [71], distribution correction estimation (DICE) [78] and fitted Q-evaluation (FQE) [40]. A brief overview of these methods can be found in Appendix C, and the specific implementations we use are documented in Appendix D. In Appendix E, we have also tested our method within a visual Q&A environment [10, 66], which follows similar mechanisms as in the two real-world experiments, *i.e.*, two types of return signals are considered though no human participants are involved.

**Baselines and Ablations.** The baselines include two variants for each of the OPE methods above, *i.e.*, (*i*) the *rescale* approach discussed in Section 2.2, and (*ii*) another variant that sets all the IHRs to be equal to the environmental rewards at corresponding steps, $r_t^{\mathcal{H}} = r_t \ \forall t \in [0, T-2]$, and then let $r_{T-1}^{\mathcal{H}} = r_{T-1} + (G_{0:T}^{\mathcal{H}} - G_{0:T})/\gamma^{T-1}$ with $G_{0:T} = \sum_t \gamma^t r_t$ being the environmental return, which is referred to as *fusion* below – this baseline may perform better when strong correlations existed between environmental and human rewards, as it intrinsically decomposes the human returns into IHRs. Conse-quently, in each experiment below, we compare the performance of the OPEHF framework extending all four types of OPE methods above, `<IS/DR/DICE/FQE>-OPEHF`, against the corresponding baselines,

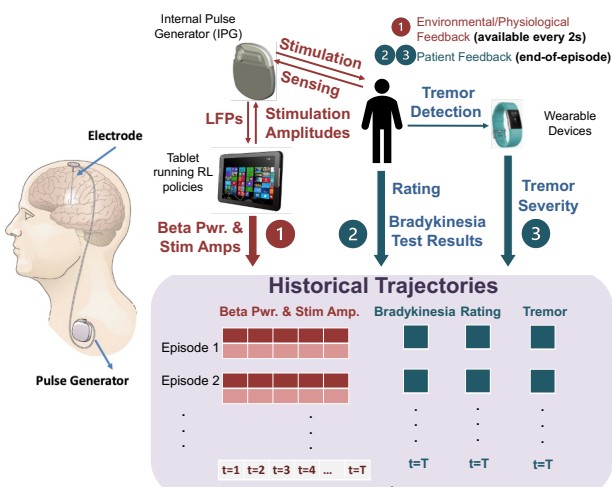

Figure 2: Setup of the neurostimulation experiments, as well as the formulation of offline trajectories. Environmental rewards and human returns are captured in streams 1 and 2-3 respectively.

`<IS/DR/DICE/FQE>-<Fusion/Rescale>`. We also include the VLM-H as an ablation baseline, as if it is a model-based approach standalone; this is achieved by sampling the estimate returns from the decoder, $\hat{G}_{0:T}^{\mathcal{H}} \sim p_\phi(G_{0:T}^{\mathcal{H}}|z_T)$.

**Metrics.** Following a recent OPE benchmark [15], three metrics are considered to validate the performance of each method, including mean absolute error (MAE), rank correlation, and regret@1. Mathematical definitions can be found in Appendix D. Also, following [15], each method is evaluated over 3 random seeds, and the mean performance (with standard errors) is reported.

### 3.1 Adaptive Neurostimulation: Deep Brain Stimulation

Adaptive neurostimulation facilitates treatments for a variety of neurological disorders [4, 11, 13, 55]. Deep brain stimulation (DBS) is a type of neurostimulation used specifically toward Parkinson's disease (PD), where an internal pulse generator (IPG), implanted under the collarbone, sends electrical stimulus to the basal ganglia (BG) area of the brain through invasive electrodes; Figure 2 illustrates

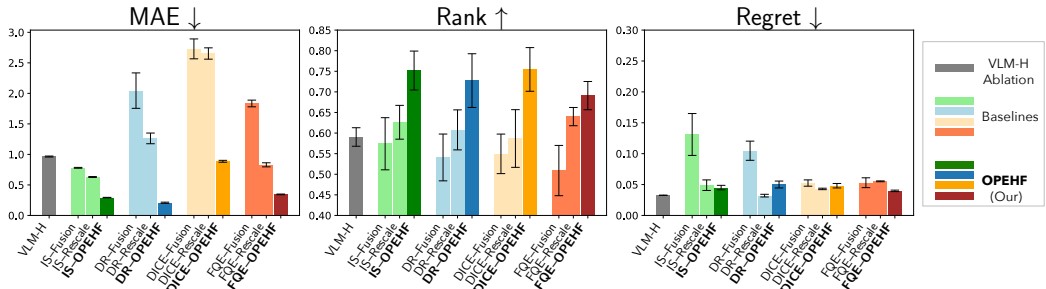

Figure 3: Results from the adaptive neurostimulation experiment, *i.e.*, deep brain stimulation (DBS). Each method is evaluated over the data collected from each patient, toward corresponding target policies, respectively. The performance shown above are averaged over all 4 human participants affected by Parkinson's disease (PD). Raw performance over each patient can be found in Appendix D.

the setup. Adaptive DBS aims to adjust the strength (amplitude) of the stimulus in real-time, to respond to irregular neuronal activities caused by PD, leveraging the local field potentials (LFPs) as the immediate feedback signals, *i.e.*, the environmental rewards. Existing works have leveraged RL for adaptive DBS over *computational* BG models [25, 20, 52, 59], using rewards defined over a physiological signal – beta-band power spectral density of LFPs (*i.e.*, the beta power) since physiologically PD could lead to increased beta power due to the irregular neuronal activations it causes [39]. However, in clinical practice, the correlation between beta power and the level of satisfaction reported by the patients varies depending on the specific characteristics of each person, as PD can cause different types of symptoms over a wide range of severity [56, 37, 5, 76]. Such findings further justify the significance of evaluating HF/human returns in the real world using OPEHF.

In this experiment, we leverage OPEHF to estimate the feedback provided by 4 *PD patients* who participate in monthly clinical testings of RL policies trained to adapt amplitudes of the stimulus toward reducing their PD symptoms, *i.e.*, bradykinesia and tremor. A mixture of behavioral policies is used to collect the offline trajectories $\rho^\beta$. Specifically, in every step, the state $s_t$ is a historical sequence of LFPs capturing neuronal activities, and the action $a_t$ up-

Table 1: Correlations between the *environmental* and *human* returns of the 6 target DBS policies associated with each PD patient.

| Patient # | *0* | *1* | *2* | *3* |
|---|---|---|---|---|
| Pearson's | -0.396 | -0.477 | -0.599 | -0.275 |
| Spearman's | -0.2 | -0.6 | 0.086 | 0.086 |

dates the amplitude of the stimulus to be sent[6]. Then, an *environmental* reward $r_t = R(s_t, a_t)$ gives a penalty if the beta power computed from the latest LFPs is greater than some threshold (to promote treatment efficacy) as well as a penalty proportional to the amplitudes of the stimulus being sent (to improve battery life of the IPG). At the end of each episode, the *human returns* $G_{0:T}^{\mathcal{H}}$ are determined from three sources (weighted by 50%, 25%, 25%, respectively), *i.e.*, (*i*) a satisfaction rating (between 1-10) provided by the patient, (*ii*) hand grasp speed as a result of the bradykinesia test [63], and (*iii*) level of tremorcalculated over the data from a wearable accelerometry [60, 6]. Each session lasts more than 10 minutes, and each discrete step above corresponds to 2 seconds in the real world; thus, the horizon $T \geq 300$ (more details are provided in Appendix D). Approval of an Institutional Review Board (IRB) is obtained from Duke University Health System, as well as the exceptional use of the DBS system by the US Food and Drug Administration (FDA).

For each patient, OPEHF and the baselines are used to estimate the human returns of 6 target policies with varied performance. The ground-truth human return for each target policy is obtained as a result of extensive clinical testing following the same schema above, over more than 100 minutes. Table 1 shows the Pearson's and Spearman's correlation coefficients [14], measuring the linear and rank correlations between the environmental returns $G_{0:T}$ and the human returns $G_{0:T}^{\mathcal{H}}$ over all the target DBS policies considered for each patient. Pearson's coefficients are all negative since the environmental reward function only issues penalties, while human returns are all captured by positive values. It can be observed that only weak-to-moderate degrees of linear correlations exist for all four patients, while ranks between $G_{0:T}$'s and $G_{0:T}^{\mathcal{H}}$'s are not preserved across patients; thus, it highlights

---

[6]RL policies only adapt the stimulation amplitudes within a safe range as determined by neurologists/neurosurgeons, making sure they will not lead to negative effects to participants.

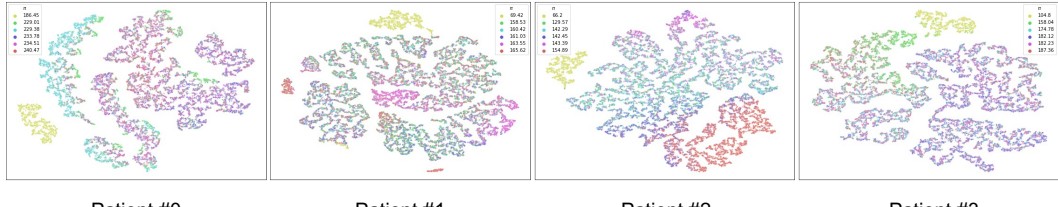

| Patient #0 | Patient #1 | Patient #2 | Patient #3 |

Figure 4: $t$-SNE visualizing the VLM-H encodings of the state-action pairs rolled out over DBS policies with different human returns (shown in the legend). It can be observed that distances among the encoded pairs associated with the policies that lead to similar returns are in general smaller, justifying the RILR objective (5).

Table 2: Results from the intelligent tutoring experiment, *i.e.*, performance achieved by our OPEHF framework compared to the ablation and baselines over all four types of downstream OPE estimators.

|  | *IS* | | | *DR* | | | *Ablation* |
|  | Fusion | Rescale | **OPEHF** (our) | Fusion | Rescale | **OPEHF** (our) | VLM-H |
|---|---|---|---|---|---|---|---|
| MAE | 0.7±0.14 | 0.77±0.08 | **0.57±0.09** | 1.03±0.07 | 1.03±0.25 | **0.86±0.04** | 1.00±0.01 |
| Rank | 0.47±0.11 | 0.4±0.09 | **0.8±0.09** | 0.33±0.05 | 0.4±0.0 | **0.53±0.2** | 0.41±0.25 |
| Regret@1 | **0.36±0.16** | **0.36±0.16** | 0.41±0.04 | 0.41±0.0 | 0.41±0.0 | 0.41±0.0 | 0.28±0.19 |

|  | *DICE* | | | *FQE* | | |
|  | Fusion | Rescale | **OPEHF** (our) | Fusion | Rescale | **OPEHF** (our) |
|---|---|---|---|---|---|---|
| MAE | 3.19±0.57 | 2.33±0.59 | **1.01±0.01** | 0.74±0.07 | 0.98±0.1 | **0.59±0.1** |
| Rank | 0.47±0.2 | 0.33±0.2 | **0.53±0.22** | 0.27±0.14 | 0.4±0.0 | **0.47±0.05** |
| Regret@1 | 0.55±0.06 | 0.45±0.18 | **0.37±0.15** | **0.36±0.16** | 0.41±0.0 | 0.41±0.0 |

the need for leveraging OPEHF to estimate human returns, which is different than the classic OPE that focus on estimating environmental returns.

The overall performance averaged across the 4-patient cohort, is reported in Fig. 3. Raw performance over every single patient can be found in Appendix D. It can be observed that our OPEHF framework significantly improves MAEs and ranks compared to the two baselines, for all 4 types of downstream OPE methods we considered (IS, DR, DICE, and FQE). Moreover, our method also significantly outperforms the ablation VLM-H in terms of these two metrics, as the VLM-H's performance is mainly determined by how well it could capture the underlying dynamics and returns. In contrast, our OPEHF framework not only leverages the latent representations learnt by the VLM-H (for regularizing RILR), it also inherits the advantages intrinsically associated with the downstream estimators; *e.g.*, low-bias nature of IS, or low-variance provided by DR. Moreover, the fusion baseline in general performs worse than the rescale baseline as expected, since no strong correlations between environmental and human returns are found, as reported in Table 1.

Note that the majority of the methods lead to similar (relatively low) regrets, as there exist a few policies that lead to human returns that are close over some patients (see the raw statistics in Appendix D). The reason is that all the policies to be extensively tested in clinics are subject to initial screening, where clinicians ensure they would not lead to undesired outcomes or pose significant risks to the patients; thus, the performance of some target policies tends to be close. Nonetheless, low MAEs and high ranks achieved by our method show that it can effectively capture the subtle differences in returns resulting from other HF signals, *i.e.*, levels of bradykinesia and tremor. Moreover, Figure 4 visualizes the VLM-H encodings over the trajectories collected from the 6 target DBS policies for each participant and shows that encoded pairs associated with the policies that lead to similar returns are in general clustered together, which justifies the importance of leveraging the similarities over latent representations to regularize the reconstruction of IHRs as in the RILR objective (5).

## 3.2 Intelligent Tutoring

Intelligent tutoring refers to a system where students can actively interact with an autonomous tutoring agent that can customize the learning content, tests, etc., to improve engagement and learning

outcomes [2, 64, 45]. OPEHF is important in such a setup for directly estimating the potential outcomes that could be obtained by students, as opposed to environmental rewards that are mostly discrete; see detailed setup below. Existing works have explored this topic over classic OPE setting in *simulations* [49, 54].

The system is deployed in an undergraduate-level introduction to probability and statistics course over 5 academic years at North Carolina State University, where the interaction logs obtained from 1,288 students who voluntarily opted-in for this experiment are recorded.[7] Specifically, each episode refers to a student working on a set of 12 problems (*i.e.*, horizon $T = 12$), where the agent suggests the student approach each problem through *independent* work, working with the *hints* provided, or directly providing the *full solution* (for studying purposes) – these options constitute the action space of the agent. The states are characterized by 140 features extracted from the logs, designed by domain experts; they include, for example, the time spent on each problem, and the correctness of the solution provided. In each step, an immediate *environmental* reward of +1 is issued if the answer submitted by students, for the current problem, is at least 80% correct (auto-graded following pre-defined rubrics). A reward of 0 is issued if the grade is less than 80% or the agent chooses the action that directly displays the full solution. Moreover, students are instructed to complete two exams, one before working on any problems and another after finishing all the problems. The normalized difference between the grades of two exams constitutes the *human* return for each episode. More details are provided in Appendix D.

The intelligent tutoring agent follows different policies across academic years, where the data collected from the first 4 years (1148 students total) constitutes the offline trajectories $\rho^\beta$ (as a result of a mixture of behavioral policies). The 4 policies deployed in the 5th year (140 students total) serve as the target policies, whose ground-truth performance is determined by averaging over the human returns of the episodes that are

Table 3: Correlations between the *environmental* and *human* returns from data collected over each academic year.

| Year # | 0 | 1 | 2 | 3 | 4 |
|---|---|---|---|---|---|
| Pearson's | 0.033 | 0.176 | 0.089 | 0.154 | 0.183 |
| Spearman's | 0.082 | 0.156 | 0.130 | 0.161 | 0.103 |

associated with each policy respectively. Table 3 documents the Pearson's and Spearman's correlation coefficients between the environmental and human returns from data collected over each academic year, showing weak linear and rank correlations across all 5 years. Such low correlations are due to the fact that the environmental rewards are discrete and do not distinguish among the agent's choices, *i.e.*, a +1 reward can be obtained either if the student works out a solution independently or by following hints, and a 0 reward is issued every time the agent chooses to display the solution even if the student could have solved the problem. As a result, such a setup makes OPEHF to be more challenging; because human returns are only available at the end of each episode, and the immediate environmental rewards do not carry substantial information toward extrapolating IHRs.

Table 2 documents the performance of OPEHF and the baselines toward estimating the human returns of the target policies. It can be observed that our OPEHF framework achieves state-of-the-art performance, over all types of downstream OPE estimators considered. This result echos the design of the VLM-H where both environmental information (state transitions and rewards) and human returns are encoded into the latent space, which helps formulate a compact and expressive latent space for regularizing the downstream RILR objective (5). Moreover, it is important to use the latent information to guide the reconstruction of IHRs (as regularizations in RILR), as opposed to using the VLM-H to predict human returns standalone; since limited convergence guarantees/error bounds can be provided for VAE-based latent models, which is illustrated in both Figure 3 and Table 2 where OPEHF largely outperforms the VLM-H ablation over MAE and rank.

## 4 Related Works

**OPE.** Majority of existing model-free OPE methods can be categorized into one of the four types, *i.e.*, IS, DR, DICE, and FQE. Recently, variants of IS and DR methods have been proposed for variance or bias reduction [34, 71, 12, 69], as well as adaptations toward unknown behavioral policies [30].

---

[7]An IRB approval is obtained from North Carolina State University. The use/test of the intelligent tutoring system is overseen by a departmental committee, ensuring it does not risk the academic performance and privacy of the participants.

DICE methods are intrinsically designed to work with offline trajectories rolled out from a mixture of behavioral policies, and existing works have introduced the DICE variants toward specific environmental setups [84, 83, 77, 78, 51, 9]. FQE extrapolates policy returns from the approximated Q-values [31, 40, 36]. There also exist model-based OPE methods [82, 18] that first captures the dynamics underlying the environment, and estimate policy performance by rolling out trajectories under the target policies. A more detailed review of existing OPE methods can be found in Appendix C. Note that these OPE methods have been designed for estimating the *environmental* returns. In contrast, the objective for OPEHF is to estimate the *human* returns which may not be strongly correlated with the environmental returns, as they are usually determined under different schemas.

**VAEs for OPE and offline RL.** There exists a long line of research developing latent models to capture the dynamics underlying environments in the context of offline RL as well as OPE. Specifically, PlaNet [27] uses recurrent neural networks to capture the transitions of latent variables over time. Latent representations learned by such VAE architectures have been used to augment the state space in offline policy optimization to improve sample efficiency, *e.g.*, in Dreamers [26, 28], SOLAR [81] and SLAC [42]. On the other hand, LatCo [65] attempts to improve sample efficiency by searching in the latent space which allows by-passing physical constraints. Also, MOPO [80], COMBO [79], and LOMPO [62] train latent models to quantify the confidence of the environmental transitions as learned from offline data and prevent the policies from following transitions over uncertain regions during policy training. Given that such models are mostly designed for improving sample efficiency in policy optimization/training, we choose to leverage the architecture from [18] for RILR as it is the first work that adapts latent models to the OPE setup.

**Reinforcement learning from human feedback (RLHF).** Recently, the concept of RLHF has been widely used in guiding RL policy optimization with the HF signals deemed more informative than the environmental rewards [8, 85, 47]. Specifically, they leverage the *ranked preference* provided by labelers to train a reward model, captured by feed-forward neural networks, that is fused with the environmental rewards to guide policy optimization. However, in this work, we focus on estimating the HF signals that serve as *direct evaluation* of the RL policies used in human-involved experiments, such as the level of satisfaction (*e.g.*, on a scale 1-10) and the treatment outcome. The reason is that in many scenarios the participants cannot revisit the same procedure multiple times, *e.g.*, patients may not undergo the same surgeries several times and rank the experiences. More importantly, OPEHF's setup is critical when online testing of RL policies may be even prohibited, without sufficient justifications over safety and efficacy upfront, as illustrated by the experiments above.

**Reward shaping.** Although reward shaping methods [3, 58, 29] pursue similar ideas of decomposing the delayed and/or sparse rewards (*e.g.*, the human return) into immediate rewards, they fundamentally rely on transforming the MDP to such that the value functions can be smoothly captured and high-return state-action pairs can be quickly identified and frequently re-visited. For example, RUDDER [3] leverages the transformed MDP that has expected future rewards equal to zero. Though the optimization objective is consistent between pre- and post-transformed MDPs, this approach likely would not converge to an optimal policy in practice. On the other hand, the performance (*i.e.,* returns) of sub-optimal policies is not preserved across the two MDPs. This significantly limits its use cases toward OPE which requires the returns resulted by sub-optimal policies to be estimated accurately. As a result, such methods are not directly applicable to the OPEHF problem we consider.

## 5   Conclusion and Future Works

Existing OPE methods fall short in estimating HF signals, as HF can be dependent upon various confounders. Thus, in this work, we introduced the OPEHF framework that revived existing OPE methods for estimating human returns, through RILR. The framework was validated over two real-world experiments and one simulation environment, outperforming the baselines in all setups. Although in the future it could be possible to extend OPEHF to facilitate estimating the HF signals needed for updating the policies similar to RLHF, we focused on policy evaluation which helped to isolate the source of improvements; as policy optimization's performance may depend on multiple factors, such as the exploration techniques used as well as the objective/optimizer chosen for updating the policy. Moreover, this work mainly focuses on the scenarios where the human returns are directly provided by the participants. So under the condition where the HF signals are provided by 3-rd parties (*e.g,* clinicians), non-trivial adaptations over this work may be needed to consider special cases such as conflicting HF signals provided by different sources.

# 6 Acknowledgements

This work is sponsored in part by the AFOSR under award number FA9550-19-1-0169, by the NIH UH3 NS103468 award, and by the NSF CNS-1652544, DUE-1726550, IIS-1651909 and DUE-2013502 awards, as well as the National AI Institute for Edge Computing Leveraging Next Generation Wireless Networks, Grant CNS-2112562. Investigational Summit RC+S systems and technical support provided by Medtronic PLC. Apple Watches were provided by Rune Labs. We thank Stephen L. Schmidt and Jennifer J. Peters from Duke University Department of Biomedical Engineering, as well as Katherine Genty from Duke University Department of Neurosurgery, for the efforts overseeing DBS experiments in the clinic.

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

## A Proofs for Section 2.2

Recall that the per-decision importance sampling (PDIS) [61] estimator follows $\hat{G}^\pi_{PDIS} = \frac{1}{N}\sum_{i=1}^{N-1}\sum_{t=0}^{T-1}\gamma^t\omega_{0:t}^{(i)}r_t^{\mathcal{H}(i)}$; here, $\omega_t^{(i)} = \frac{\pi(a_t^{(i)}|s_t^{(i)})}{\hat{\beta}(a_t^{(i)}|s_t^{(i)})}$, and henceforth $\omega_{0:t}^{(i)} = \prod_{k=0}^{t}\frac{\pi(a_k^{(i)}|s_k^{(i)})}{\hat{\beta}(a_k^{(i)}|s_k^{(i)})}$ are the PDIS weights for offline trajectory $\tau^{(i)}$. To simplify our notation, we consider the PDIS estimator defined over a single trajectory, *i.e.*,

$$\hat{G}^\pi_{PDIS} = \sum_{t=0}^{T-1}\gamma^t\omega_{0:t}r_t^{\mathcal{H}}, \tag{6}$$

as the results can be carried to $N$ trajectories by multiplying with a factor $1/N$. We omit the superscript-ed $^{(i)}$'s in the rest of the proof also for conciseness.

### A.1 Proof of Proposition 1

*Proof.* We start with a lemma from [46].

**Lemma 1** ([46]). *Given $X_t$ and $Y_t$ as two sequences of random variables. Then*

$$2\sum_{t<k}\mathbb{E}[Y_tY_k] - 2\sum_{t<k}\mathbb{E}\Big[\mathbb{E}[Y_t|X_t]\mathbb{E}[Y_k|X_k]\Big] \leq \mathbb{V}\Big(\sum_t Y_t\Big) - \mathbb{V}\Big(\sum_t\mathbb{E}[Y_t|X_t]\Big), \tag{7}$$

*where $\mathbb{V}(\cdot)$ refer to the variances.*

Now, if we set $Y_t = r_t^{\mathcal{H}}\omega_{0:T}$ and $X_t = \tau_{0:t}$ which is a segment (from $t=0$ to $t=t$) of an offline trajectory $\tau$, it is sufficient to prove Proposition 1 by proving that for any $1 \leq t+1 \leq k \leq T$,

$$\mathbb{E}[r_t^{\mathcal{H}}r_k^{\mathcal{H}}\omega_{0:t}\omega_{0:k}] \leq \mathbb{E}[r_t^{\mathcal{H}}r_k^{\mathcal{H}}\omega_{0:T-1}\omega_{0:T-1}]; \tag{8}$$

since

$$\mathbb{E}[r_t^{\mathcal{H}}r_k^{\mathcal{H}}\omega_{0:t}\omega_{0:k}] = \mathbb{E}[\mathbb{E}[Y_t|X_t]\mathbb{E}[Y_k|X_k]] \tag{9}$$
$$\leq \mathbb{E}[Y_tY_k] \tag{10}$$
$$= \mathbb{E}[r_t^{\mathcal{H}}r_k^{\mathcal{H}}\omega_{0:T-1}\omega_{0:T-1}]. \tag{11}$$

Applying the law of total expectations to (8), *i.e.*,

$$\mathbb{E}\big[\mathbb{E}[r_t^{\mathcal{H}}r_k^{\mathcal{H}}\omega_{0:t}\omega_{0:k}|\tau_{0:t}]\big] \leq \mathbb{E}\big[\mathbb{E}[r_t^{\mathcal{H}}r_k^{\mathcal{H}}\omega_{0:T-1}\omega_{0:T-1}|\tau_{0:t}]\big]; \tag{12}$$

then it is sufficient to show

$$\mathbb{E}[r_t^{\mathcal{H}}r_k^{\mathcal{H}}\omega_{0:t}\omega_{0:k}|\tau_{0:t}] \leq \mathbb{E}[r_t^{\mathcal{H}}r_k^{\mathcal{H}}\omega_{0:T-1}\omega_{0:T-1}|\tau_{0:t}]. \tag{13}$$

We start by showing that

$$\mathbb{E}[r_t^{\mathcal{H}}r_k^{\mathcal{H}}\omega_{0:t}\omega_{0:k}|\tau_{0:t}] = \omega_{0:t}^2\mathbb{E}[r_t^{\mathcal{H}}|\tau_{0:t}]\mathbb{E}[r_k^{\mathcal{H}}\omega_{t+1:k}|\tau_{0:t}] \tag{14}$$
$$= \omega_{0:t}^2\mathbb{E}[r_t^{\mathcal{H}}|\tau_{0:t}]\mathbb{E}[r_k^{\mathcal{H}}\omega_{t+1:T-1}|\tau_{0:t}]\mathbb{E}[\omega_{t+1:T-1}|\tau_{0:t}]. \tag{15}$$

The transition above follows from the fact that the likelihood ratio $\omega_{0:t}$ is a martingale as shown in [41, 46], *i.e.*, $\mathbb{E}[\omega_{0:T-1}|\tau_{0:t}] = \omega_{0:t}$, which implies $\mathbb{E}[\omega_{t+1:T-1}] = 1$ and therefore

$$\mathbb{E}[r_k^{\mathcal{H}}\omega_{t+1:k}|\tau_{0:t}] \tag{16}$$
$$= \mathbb{E}[r_k^{\mathcal{H}}\omega_{t+1:k}|\tau_{0:t}]\mathbb{E}[\omega_{k+1:T-1}|\tau_{0:t}] \tag{17}$$
$$= \mathbb{E}[r_k^{\mathcal{H}}\omega_{t+1:T-1}|\tau_{0:t}]. \tag{18}$$

Note that we keep using the notation $\mathbb{E}[r_t^{\mathcal{H}}|\tau_{0:t}]$ in (14) and (15), due to the setting of HMDP that $r_t^{\mathcal{H}}$ is a variable sampled from the distribution $R^{\mathcal{H}}(\cdot|s_t, a_t)$ given $(s_t, a_t)$; see Section 2.1.

Since $\tau_{0:t}$ is given, $r_k^{\mathcal{H}}$ and $\omega_{t+1:T-1}$ are equivalent to $r_{k-t-1}^{(j)}$ and $\omega_{0:T-t-2}^{(j)}$ for some other trajectory $\tau^{(j)} \sim \rho^{\beta}$. Also given the statement that $\omega_{0:T-t-2}^{(j)}$ and $r_{k-t-1}^{(j)}\omega_{0:T-t-2}^{(j)}$ are positively correlated, it follows from (15) that

$$\mathbb{E}[r_t^{\mathcal{H}} r_k^{\mathcal{H}} \omega_{0:t}\omega_{0:k}|\tau_{0:t}] \tag{19}$$

$$=\omega_{0:t}^2 \mathbb{E}[r_t^{\mathcal{H}}|\tau_{0:t}]\mathbb{E}[r_k^{\mathcal{H}}\omega_{t+1:T-1}|\tau_{0:t}]\mathbb{E}[\omega_{t+1:T-1}|\tau_{0:t}] \tag{20}$$

$$\leq\omega_{0:t}^2 \mathbb{E}[r_t^{\mathcal{H}}|\tau_{0:t}]\mathbb{E}[r_k^{\mathcal{H}}\omega_{t+1:T-1}\omega_{t+1:T-1}|\tau_{0:t}] \tag{21}$$

$$=\mathbb{E}[r_t^{\mathcal{H}} r_k^{\mathcal{H}} \omega_{0:T-1}\omega_{0:T-1}|\tau_{0:t}], \tag{22}$$

which justifies (13) and completes the proof. $\qquad\square$

## B  Derivation of the ELBO (2)

Note that unlike the ELBO in [18], the VLM-H includes an additional component that estimates the human return $G_{0:T}^{\mathcal{H}}$ of each trajectory, *i.e.*,

$$\log p_\phi(s_{0:T}, r_{0:T-1}, G_{0:T}^{\mathcal{H}}) \tag{23}$$

$$= \log \int_{z_{1:T}\in\mathcal{Z}} p_\phi(s_{0:T}, z_{1:T}, r_{0:T-1}, G_{0:T}^{\mathcal{H}})dz \tag{24}$$

$$= \log \int_{z_{1:T}\in\mathcal{Z}} \frac{p_\phi(s_{0:T}, z_{1:T}, r_{0:T-1}, G_{0:T}^{\mathcal{H}})}{q_\psi(z_{0:T}|s_{0:T}, a_{0:T-1})} q_\psi(z_{0:T}|s_{0:T}, a_{0:T-1})dz \tag{25}$$

$$\geq\mathbb{E}_{q_\psi}[\log p(z_0) + \log p_\phi(s_{0:T}, z_{1:T}, r_{0:T-1}|z_0) + \log p_\phi(G_{0:T}^{\mathcal{H}}|z_T) - \log q_\psi(z_{0:T}|s_{0:T}, a_{0:T-1})] \tag{26}$$

$$=\mathbb{E}_{q_\psi}\Big[\log p(z_0) + \log p_\phi(s_0|z_0) + \log p_\phi(G_{0:T}^{\mathcal{H}}|z_T) + \sum_{t=1}^{T}\log p_\phi(s_t, z_t, r_{t-1}|z_{t-1}, a_{t-1})$$
$$- \log q_\psi(z_0|s_0) - \sum_{t=1}^{T}\log q_\psi(z_t|z_{t-1}, a_{t-1}, s_t)\Big] \tag{27}$$

$$=\mathbb{E}_{q_\psi}\Big[\log p(z_0) - \log q_\psi(z_0|s_0) + \log p_\phi(s_0|z_0) + \log p_\phi(G_{0:T}^{\mathcal{H}}|z_T) - \sum_{t=1}^{T}\log q_\psi(z_t|z_{t-1}, a_{t-1}, s_t)$$
$$+ \sum_{t=1}^{T}\log\big(p_\phi(s_t|z_t)p_\phi(r_{t-1}|z_t)p_\phi(z_t|z_{t-1}, a_{t-1})\big)\Big] \tag{28}$$

$$=\mathbb{E}_{q_\psi}\Big[\log p_\phi(G_{0:T}^{\mathcal{H}}|z_T) + \sum_{t=0}^{T}\log p_\phi(s_t|z_t) + \sum_{t=1}^{T}\log p_\phi(r_{t-1}|z_t)$$
$$- KL\big(q_\psi(z_0|s_0)||p(z_0)\big) - \sum_{t=1}^{T}KL\big(q_\psi(z_t|z_{t-1}, a_{t-1}, s_t)||p_\phi(z_t|z_{t-1}, a_{t-1})\big)\Big]. \tag{29}$$

Note that to simplify our presentation, we omit $\phi, \psi, \rho^\beta$ as part of the conditional terms in the joint likelihoods. The transition from (25) to (26) follows Jensen's inequality.

## C  Brief Overview of Existing OPE methods

### C.1  Importance Sampling (IS) and per-decision IS (PDIS)

IS refers to a statistical technique that can calculate the expectation of a function $f(x)$ w.r.t. an unknown distribution $p(x)$ using a given distribution $q(x)$ through re-weighting, *i.e.*,

$$\mathbb{E}_p[f(x)] = \mathbb{E}_q\left[\frac{f(x)p(x)}{q(x)}\right]. \tag{30}$$

This technique can be applied in the context of OPE by setting $f(x)$ as the accumulated return $G_{0:T}$, $p$ as the trajectory distribution $\rho^\pi$ over the target policy $\pi$, and $q$ as the trajectory distribution $\rho^\beta$ over the behavioral policy $\beta$.

According to [61], the vanilla IS estimator follows

$$\hat{G}_{IS}^{\pi} = \frac{1}{N} \sum_{i=1}^{N-1} \omega_{0:T-1}^{(i)} G_{0:T}^{(i)},$$ (31)

where $\omega_{0:T-1}^{(i)} = \prod_{t=0}^{T-1} \frac{\pi(a_t|s_t)}{\beta(a_t|s_t)}$ refers to the IS weight, and $G_{0:T}^{(i)}$ is the return for the $i$-th (out of $N$) offline trajectory. On the other hand, the PDIS estimator follows

$$\hat{G}_{PDIS}^{\pi} = \frac{1}{N} \sum_{i=1}^{N-1} \sum_{t=0}^{T-1} \gamma^t \omega_{0:t}^{(i)} r_t^{(i)},$$ (32)

with $\omega_{0:t}^{(i)} = \prod_{k=0}^{t} \frac{\pi(a_k|s_k)}{\beta(a_k|s_k)}$ being the PDIS weight, $r_t^{(i)}$ is the environmental reward obtained at the $t$-th step of the $i$-th offline trajectory, and $\gamma$ is the discounting factor.

## C.2 Doubly Robust (DR)

DR attempts to reduce the variance of IS estimation by introducing the value function approximation, which trades off estimation variance by making the estimation biased. The non-recursive definition of DR estimation is provided in [71], *i.e.*,

$$\hat{G}_{DR}^{\pi} = \frac{1}{N} \sum_{i=1}^{N-1} \sum_{t=0}^{T-1} \gamma^t \omega_{0:t}^{(i)} r_t^{(i)} - \frac{1}{N} \sum_{i=1}^{N-1} \sum_{t=0}^{T-1} \left( \gamma^t \omega_{0:t}^{(i)} r_t^{(i)} Q^{\pi}(s_t^{(i)}, a_t^{(i)}) - \omega_{0:t-1}^{(i)} V^{\pi}(s_t^{(i)}) \right);$$ (33)

here, $Q^{\pi}(\cdot, \cdot)$ and $V^{\pi}(\cdot)$ are the Q-function and value function, respectively, over the target policy, while $s_t^{(i)}$ and $a_t^{(i)}$ are the state and action taken at the $t$-th step of the $i$-th offline trajectory.

## C.3 DIstributional Correction Estimation (DICE)

DICE aims to estimate the propensity of the target policy to visit particular state-action pairs relative to their likelihood of appearing in the offline trajectories, *i.e.*,

$$\hat{G}_{DICE}^{\pi} = \mathbb{E}_{(s,a,r)\sim\rho^{\beta}} \left[ \frac{d^{\pi}(s,a)}{d^{\beta}(s,a)} \cdot r \right],$$ (34)

where $\frac{d^{\pi}(s,a)}{d^{\beta}(s,a)}$ is the distribution correction ratio. Existing DICE variants [84, 83, 78, 51, 9] seek to approximate the ratio without knowledge of $d^{\pi}$ or $d^{\beta}$, while a recent work has summarized that all existing variants can be formulated as regularized Lagrangians of the same linear program [78], unifying the choice of regularizations and constraints used in their original objectives.

## C.4 Fitted Q-Evaluation (FQE)

According to [40], FQE approximates the Q-function over the target policy by minimizing the loss

$$\min_{\kappa} \mathbb{E}_{(s_t,a_t,r_t)\sim\rho^{\beta}} \left[ \left( Q(s_t, a_t; \kappa) - y_t \right)^2 \right],$$ (35)

$$\text{s.t.} \quad y_t = r_t + \gamma Q(s_{t+1}, \pi(s_{t+1}); \kappa);$$ (36)

here, the approximated Q-function is parameterized by $\kappa$. Note that it is different from the objective for classic Q-learning, or fitted Q-iteration, by using $\pi(s_{t+1})$ instead of $\max_a Q(s_{t+1}, a; \kappa)$ in the target $y_t$.

# D  Additional Experimental Details and Results

In this section we provide additional details that are supplement to the discussions in Section 3.

### D.1 Definition of the OPE Metrics

**Mean Absolute error (MAE).** MAE is defined as the absolute difference between the actual return and estimated return of a policy: $MAE = |V^\pi - \hat{V}^\pi|$; here, $V^\pi$ is the actual value of the policy $\pi$, and $\hat{V}^\pi$ is the estimated value of $\pi$.

**Rank correlation.** Rank correlation measures the Spearman's rank correlation coefficient between the ordinal rankings of the estimated returns and actual returns across policies, *i.e.*,
$\rho = \frac{Cov(\text{rank}(V^\pi_{1:P}), \text{rank}(\hat{V}^\pi_{1:P}))}{\sigma(\text{rank}(V^\pi_{1:P}))\sigma(\text{rank}(\hat{V}^\pi_{1:P}))}$, where $\text{rank}(V^\pi_{1:P})$ is the ordinal rankings of the actual returns, and $\text{rank}(\hat{V}^\pi_{1:P})$ is the ordinal rankings of the OPE-estimated returns.

**Regret@1.** Regret@1 is the (normalized) difference between the value of the actual best policy, against the value of the policy associated with the best OPE-estimated return, which is defined as $(\max_{i \in 1:P} V^\pi_i - \max_{j \in \text{best}(1:P)} V^\pi_j)/\max_{i \in 1:P} V^\pi_i$, where $\text{best}(1:P)$ denotes the index of the best policy over the set of $P$ policies as measured by estimated values $\hat{V}^\pi$.

### D.2 Implementations and Hyper-parameters

**VLM-H.** Both encoder $q_\psi(z_t|z_{t-1}, a_{t-1}, s_t)$ and decoder $p_\phi(z_t|z_{t-1}, a_{t-1})$ of VLM-H are configured to be LSTMs with 64 nodes for the hidden states, followed by two fully connected layers with 128 and 64 nodes each. All other parts of the encoder and decoder, including $q_\psi(z_0|s_0), p_\phi(G^{\mathcal{H}}_{0:T}|z_T), p_\phi(s_t|z_t), p_\phi(r_{t-1}|z_t)$, are multi-layered perceptrons (MLPs) with two fully connected layers with 128 and 64 nodes each. The regularization term introduced in [18] is added to the ELBO (2) with a scaling factor $C'$ balancing the scale of the two terms, which is applied over the LSTM states between $q_\psi(z_t|z_{t-1}, a_{t-1}, s_t)$ and $p_\phi(z_t|z_{t-1}, a_{t-1})$. The scale of this regularization is selected from $C' = \{$1e-04, 5e-03, 1e-03, 5e-02, 1e-02, 0.1, 1., 2., 5.$\}$. Learning rate is tuned by grid search from $\{$0.003, 0.001, 0.0007, 0.0005, 0.0003, 0.0001, 0.00005$\}$. Exponential decay is applied to the learning rate, which decays the learning rate by 0.997 every iteration. The total number of training epochs is set to be 20 and minibatch size set to 64. Adam optimizer is used to perform gradient descent. $L_2$ weight decay with coefficient 0.001 and batch normalization are applied to all hidden fully connected layers. The top-10 models/checkpoints achieving the highest ELBOs proceed to the RILR step.

**Downstream OPE Estimators.** The implementations of downstream OPE estimators are built on top of [36][8], which provides the functionalities for using per-decision IS with behavioral policy estimation [30], DR [71], DICE [78] or FQE [40] to process the trajectories with reconstructed IHRs and estimate human returns.

**Code Availabilities and Computational Resources.** Code implementations for all three components above can be found in the supplementary materials attached. All experimental workloads are distributed among 4 Nvidia RTX A5000 24GB and 3 Nvidia Quadro RTX 6000 24GB graphics cards.

### D.3 More on the Adaptive Neurostimulation Experiment – Deep Brain Stimulation (DBS)

**DBS Hardware and Implementation.** The internal pulse generator (IPG) is part of the Medtronic's Summit RC+S system. Moreover, Summit software APIs are provided by Medtronic to be leveraged to implement the RL policies to be tested in the clinic. The overall pipeline is shown in Figure 5. Specifically, each control cycle (an MDP step) takes 2 seconds, where the IPG collects local field potentials (LFPs) from the basal ganglia (BG) region of the participant's brain at a sampling rate of 500 Hz. Then, the LFPs are sent to the research tablet for signal processing, after which the beta-band power spectral densities (beta powers) are calculated and used to constitute the MDP state and reward signals. The beta powers are calculated by applying fast Fourior transform (FFT), with 512 bin width, to the LFPs collected in the past 2 seconds. The beta-band usually refers to the frequency band of 13-35 Hz, however, clinicians and electrical engineers had customized the range for each specific

---

[8]Original implementation obtained from `https://github.com/google-research/google-research/tree/master/policy_eval`, following the Apache License v2.0

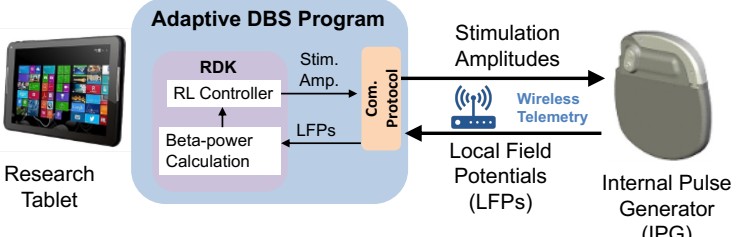

Figure 5: Implementation of RL control on the research tablet over the API provided by the manufacturer of the DBS device. During clinical testing, the research tablet evaluates the RL policies which determine the stimulation amplitudes to be used in each step (every 2 seconds). Then, by communicating with the internal pulse generator (IPG) through wireless telemetry, the IPG adjust the stimulus accordingly and sends back the local field potentials (LFPs) which are used to determine the MDP state and reward for the next step.

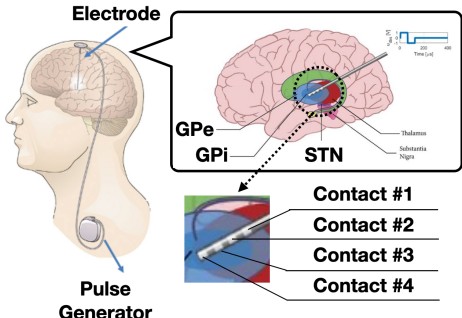

Figure 6: An illustration towards the IPG electrodes are placed in the basal ganglia (BG) region of the brain – 4-contact electrodes are placed in the subthalmic nucleus (STN) and globus pallidus (GP).

patient, in order to allow beta powers to better capture the irregular neuronal behaviors. The specific range for each patient can be found in the paragraph below (*i.e.*, patient characteristics). According to the latest LFP and beta power signals, the RL policy adjusts the amplitude of the electrical stimulus to be used for the next cycle, which is then sent back to the IPG through wireless telemetry. In the BG region of participant's brain, as illustrated in Fig. 6, 4-contact electrodes are placed in the subthalmic nucleus (STN) and globus pallidus (GP) regions. Monopolar stimulation was delivered on a single contact on each lead (with the case serving as counter-electrode). The two contacts surrounding the stimulation contact were used for sensing LFPs (*i.e.*, sandwich sensing).

The IPG device was already approved for clinical research use by the FDA. The changes made to the software program were also approved by the FDA via an investigational device exemption (IDE). Corresponding IRB approvals were obtained from Duke University Health System. All participants were compensated as per institutional and administrative guidelines.

**Patient Characteristics.** As discussed in Section 3.1, all PD participants have unique characteristics toward varied PD severity and symptoms, which can be found in [22].

**The MDP Setup.** The state and action space, as well as rewards are briefly introduced in Section 3.1. Here we provide the details that are omitted in the main paper. Each discrete step in MDP corresponds to a control cycle which lasts 2 seconds; see the *DBS hardware and implementation* paragraph above. Each episode lasts at least $T = 300$ steps (*i.e.*, the horizon), or more than 10 minutes in the real world. Specifically, states are constituted by a historical sequence of beta powers obtained from past 10 steps, following a similar setup as in [20, 52]. The actions $a \in \mathcal{A}$ are in $[\xi, 100\%] \subset \mathbb{R}, 0\% \leq \xi < 100\%$, which represent the percentage over the maximum amplitude of the stimulus the IPG can employ (determined by clinicians). Specifically, $\xi = 40\%$ for patient #0 and $\xi = 60\%$ for patients #1-#3, to ensure safety and minimum efficacy of DBS during clinical testing. The environmental rewards are

Table 4: Raw results of the adaptive neurostimulation experiment from each patient (**Patient #0**).

| | IS | | | DR | | | Ablation |
|---|---|---|---|---|---|---|---|
| | Fusion | Rescale | **OPEHF** (our) | Fusion | Rescale | **OPEHF** (our) | VLM-H |
| MAE | 0.96±0.0 | 0.74±0.0 | **0.17±0.01** | 2.21±0.38 | 1.65±0.07 | **0.09±0.01** | 0.97±0.02 |
| Rank | 0.65±0.01 | 0.65±0.01 | **0.74±0.07** | 0.43±0.05 | **0.63±0.08** | 0.58±0.1 | 0.73±0.09 |
| Regret@1 | **0.05 ± 0.0** | **0.05 ± 0.0** | **0.04 ± 0.01** | 0.03 ± 0.38 | 0.02 ± 0.07 | 0.03 ± 0.01 | 0.02±0.02 |

| | DICE | | | FQE | | |
|---|---|---|---|---|---|---|
| | Fusion | Rescale | **OPEHF** (our) | Fusion | Rescale | **OPEHF** (our) |
| MAE | 4.59±0.17 | 4.0±0.2 | **0.98±0.01** | 1.64±0.03 | 0.63±0.0 | **0.27±0.0** |
| Rank | 0.52±0.05 | 0.56±0.08 | **0.62±0.09** | 0.48±0.07 | 0.49±0.02 | **0.58±0.07** |
| Regret@1 | 0.05 ± 0.17 | 0.02 ± 0.2 | **0.03 ± 0.01** | **0.02 ± 0.03** | 0.05 ± 0.0 | **0.03 ± 0.0** |

Table 5: Raw results of the adaptive neurostimulation experiment from each patient (**Patient #1**).

| | IS | | | DR | | | Ablation |
|---|---|---|---|---|---|---|---|
| | Fusion | Rescale | **OPEHF** (our) | Fusion | Rescale | **OPEHF** (our) | VLM-H |
| MAE | 0.59±0.0 | 0.44±0.0 | **0.43±0.01** | 2.11±0.14 | 1.03±0.08 | **0.27±0.03** | 0.98±0.01 |
| Rank | 0.58±0.06 | 0.52±0.08 | **0.66±0.08** | 0.74±0.07 | 0.6±0.05 | **0.87±0.02** | 0.58±0.03 |
| Regret@1 | 0.21±0.0 | **0.02±0.0** | **0.02±0.01** | 0.01±0.14 | **0.0±0.08** | 0.02±0.03 | 0.03±0.01 |

| | DICE | | | FQE | | |
|---|---|---|---|---|---|---|
| | Fusion | Rescale | **OPEHF** (our) | Fusion | Rescale | **OPEHF** (our) |
| MAE | 0.47±0.01 | **0.45±0.01** | 0.87±0.01 | 2.53±0.07 | 1.05±0.09 | **0.53±0.02** |
| Rank | 0.73±0.05 | 0.72±0.09 | **0.9±0.03** | 0.48±0.12 | **0.76±0.01** | 0.74±0.06 |
| Regret@1 | **0.03±0.01** | **0.03±0.01** | **0.03±0.01** | **0.01±0.07** | 0.04±0.09 | **0.02±0.02** |

defined following, *i.e.*,

$$R(s, a, s') = \begin{cases} -1 - 0.2a & \text{if latest beta power is above a threshold,} \\ -0.2a & \text{if latest beta power is below a threshold;} \end{cases} \tag{37}$$

here, $-0.2a$ is the penalty for stimulating with higher-than-needed amplitudes, to preserve the runtime of the IPG device (powered by a rechargeable battery) between recharges, and an additional $-1$ is given if stimulating with the amplitude determined by the policy cannot reduce beta power to below a threshold specific to each patient. The thresholds above are set to be the lower 20% quantile of the beta powers observed from initial exploration (introduced below). The human return at the end of each episode is quantified as a 50%-25%-25% weighted sum over the patient rating, hand grasp speed captured from performing the maneuver (rapid and full extension and close of all fingers) for the bradykinesia test, as well as the proportional length of the session where the patient displays tremor (as captured by a wearable accelerometry).

**Collection of Offline Trajectories.** Mixed types of controllers are used for initial exploration of the environment (for obtaining target policies introduced below), and the resulting trajectories are used as the offline training data for OPEHF. Specifically, a random controller that uniformly sample the stimulation amplitudes from $[\xi, 100\%]$, and a proportional-integral (PI) controller tuned by clinicians and electrical engineers, are used at the initial phase of clinical testing. They help identify the state-action regions that can lead to non-trivial returns. Then, such data are used to train RL policies – following deep deterministic policy gradient (DDPG), the deterministic counterpart of the soft actor-critic (SAC) – for initial exploration, most of which can only obtain up to 60% returns as obtained by the target policies below *on average*. This makes OPEHF challenging considering the limited exploration coverage provided by the offline trajectories.

**Target Policies.** A total of 6 target policies, to be evaluated by OPEHF, are considered for each patient. One of them is an open-loop policy that always stimulate with the lowest possible amplitude

Table 6: Raw results of the adaptive neurostimulation experiment from each patient (**Patient #2**).

| | IS | | | DR | | | Ablation |
| | Fusion | Rescale | **OPEHF** (our) | Fusion | Rescale | **OPEHF** (our) | *VLM-H* |
|---|---|---|---|---|---|---|---|
| MAE | 0.63±0.03 | 0.7±0.02 | **0.33±0.01** | 2.17±0.3 | 1.11±0.16 | **0.3±0.01** | *0.97±0.02* |
| Rank | 0.65±0.11 | 0.71±0.05 | **0.9±0.03** | 0.52±0.09 | 0.59±0.04 | **0.84±0.03** | *0.6±0.0* |
| Regret@1 | **0.05±0.03** | **0.08±0.02** | **0.08±0.01** | **0.03±0.3** | 0.08±0.16 | 0.1±0.01 | *0±0.02* |

| | DICE | | | FQE | | |
| | Fusion | Rescale | **OPEHF** (our) | Fusion | Rescale | **OPEHF** (our) |
|---|---|---|---|---|---|---|
| MAE | 5.44±0.55 | 3.52±0.12 | **0.66±0.02** | 1.6±0.06 | 0.81±0.02 | **0.38±0.01** |
| Rank | 0.66±0.09 | 0.48±0.08 | **0.85±0.05** | 0.6±0.05 | 0.67±0.04 | **0.93±0.01** |
| Regret@1 | 0.11±0.55 | **0.08±0.12** | 0.1±0.02 | 0.08±0.06 | **0.08±0.02** | **0.08±0.01** |

Table 7: Raw results of the adaptive neurostimulation experiment from each patient (**Patient #3**).

| | IS | | | DR | | | Ablation |
| | Fusion | Rescale | **OPEHF** (our) | Fusion | Rescale | **OPEHF** (our) | *VLM-H* |
|---|---|---|---|---|---|---|---|
| MAE | 0.94±0.01 | 0.76±0.01 | **0.24±0.0** | 1.69±0.51 | 0.69±0.06 | **0.17±0.01** | *0.94±0.01* |
| Rank | 0.42±0.11 | 0.16±0.03 | **0.71±0.05** | 0.48±0.05 | 0.42±0.11 | **0.62±0.14** | *0.45±0.06* |
| Regret@1 | 0.21±0.01 | **0.01±0.01** | 0.04±0.0 | 0.35±0.51 | 0.1±0.06 | **0.04±0.01** | *0.03±0.01* |

| | DICE | | | FQE | | |
| | Fusion | Rescale | **OPEHF** (our) | Fusion | Rescale | **OPEHF** (our) |
|---|---|---|---|---|---|---|
| MAE | **0.41±0.02** | 1.15±0.07 | 1.03±0.04 | 1.56±0.09 | 0.53±0.02 | **0.19±0.01** |
| Rank | 0.28±0.05 | 0.22±0.03 | **0.66±0.07** | 0.48±0.05 | 0.29±0.02 | **0.52±0.01** |
| Regret@1 | **0.02±0.02** | 0.03±0.07 | **0.03±0.04** | 0.1±0.09 | 0.05±0.02 | **0.03±0.01** |

($\xi\%$), benchmarking the worst-case control performance. The other 5 target DBS policies are all trained using DDPG. Specifically, each is trained over a growing experience buffer containing the data collected from the latest trials, in addition to the fixed set of offline trajectories above; this would lead to a set of target policies with varied performance, as the policies obtained in the later stage tend to be more optimal since the environment has been explored to a broader extent, compared to the policies obtained at earlier stages. Moreover, the policies have to demonstrate at least moderate control efficacy (as determined by clinicians), in order to be considered as target policies; since the ground-truth return of each target policy is determined by averaging the human returns over more than 10 sessions of clinical testing ($> 100$ minutes in total). The raw human returns of all target policies over each patient can be found in Tables 8 to 11, as well as in the legends of Figure 4. Note that, for each patient, there may exist a few policies that lead to close returns, due to the testing limitations introduced above, *i.e.*, the use of $\xi$ as well as the safety protocol for long-term testing (introduced above).

**Correlation between beta power and patient feedback.** As discussed in Section 3.1, existing research [56, 37, 5, 76] have found inconsistency between the beta power and patient feedback, due to confounders related to patient's specific characteristics as well as varied symptoms caused by PD. Figure 7 also shows that only low-to-moderate correlations can be found between beta power and the HF signals that are used to define the human returns in general. This makes OPEHF more challenging as the environmental signals may contain limited information useful for extrapolating human returns.

### D.4 More on the Intelligent Tutoring Experiment

**Experimental Setup.** All the students follow a 4-step procedure, *i.e.*, (*i*) study with textbook, (*ii*) pre-test, (*iii*) study with the intelligent tutoring system, (*iv*) post-test. The RL policies are deployed only in step (*iii*) following the MDP setup introduced in Section 3.2. The human returns are quantified

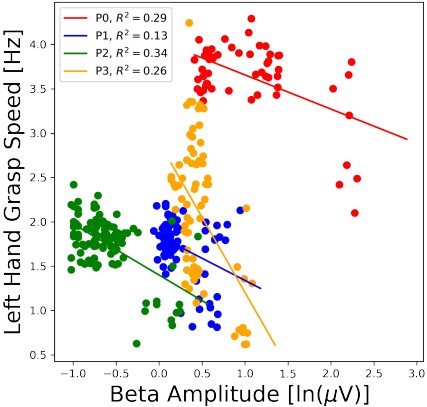

(a) Low correlations are found between beta power and the left hand grasp speeds as result of the bradykinesia test.

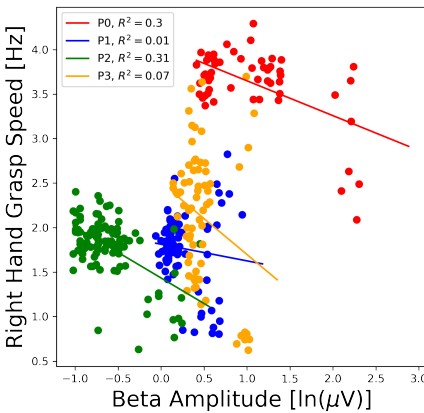

(b) Low correlations are found between beta power and the right hand grasp speeds as result of the bradykinesia test.

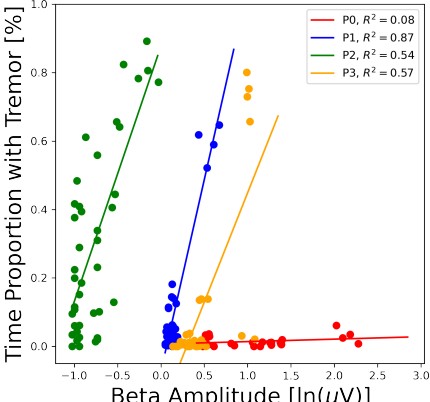

(c) Low-to-moderator correlations over 3 patients, and strong correlation over 1 patient, are found between beta power and the tremor severity quantified as the total time (proportional to the horizon) when patients experience tremor (captured by wearable accelerometry) in each testing session.

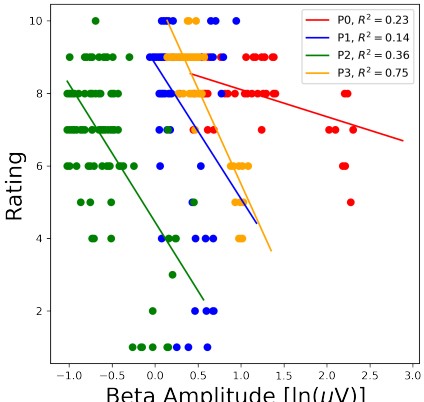

(d) Low-to-moderator correlations over 3 patients, and strong correlation over 1 patient, are found between beta power and the rating (1-10) provided by the patients at the end of each testing session.

Figure 7: Correlation between the beta power and HF provided by patients from all cinical sessions.

as the normalized gain over the difference between the scores from post-test and pre-test, *i.e.*,

$$G_{0:T}^{\mathcal{H}} = \frac{score_{post} - score_{pre}}{\sqrt{1 - score_{pre}}}. \tag{38}$$

Specifically, pre-test consists of a total of 14 single- and multiple-principle problems. No feedback are given following the pre-test, nor are they allowed to re-visit any questions from the test. After working with the intelligent tutoring system, the students take the post-test which contain 20 problems, 14 of which are isomorphic to the pre-test and the remaining ones are non-isomorphic multiple-principle problems. Both tests were auto-graded following a specific rubric.

**More on the MDP States.** As introduced in Section 3.2, the states are constituted by 140 expert-selected features; each fall into one of the five categories including (*i*) autonomy (10 features),

Table 8: Environmental and human returns of all the target policies pertaining to **Patient #0**, from the adaptive neurostimulation experiment.

| Target Policy | Environmental Returns | Human Returns (x10) |
|:---:|:---:|:---:|
| #0 | -64.18 | 182.12 |
| #1 | -128.96 | 187.36 |
| #2 | -131.37 | 174.78 |
| #3 | -134.84 | 182.23 |
| #4 | -129.82 | 158.04 |
| #5 | -80.69 | 104.8 |

Table 9: Environmental and human returns of all the target policies pertaining to **Patient #1**, from the adaptive neurostimulation experiment.

| Target Policy | Environmental Returns | Human Returns (x10) |
|:---:|:---:|:---:|
| #0 | -144.51 | 240.47 |
| #1 | -155.0 | 234.51 |
| #2 | -156.53 | 233.78 |
| #3 | -135.75 | 229.01 |
| #4 | -47.19 | 229.38 |
| #5 | -86.13 | 186.45 |

capturing the effort spent on solving problems in the intelligent tutoring system, such as the number of times the student revisit a problem, (*ii*) temporal information (29 features), including the time spent on each problem etc., (*iii*) problem-specific (35 features), such as the difficulty of each problem, (*iv*) performance (57 features), such as the grade students received on each problem, (*v*) hint-related (11 features), including the total amount of hints requested, if applicable.

**Target Policies.** A total of four target policies are considered. One of the policies randomly selects an action at each state, which benchmarks the worst-case performance. The other 3 policies are trained using deep Q-network (DQN) with different learning rates, from {1e-3, 1e-4, 1e-5}.

# E   A Challenging Simulation Environment: Visual Q&A Dialogue

We also consider the visual Q&A dialogue environment [10], following the recent framework [66] that generalizes offline RL towards language generation tasks. In this environment, language generation agents are trained to ask questions over the image captions that are given (without showing the actual images), in order to gain as much information from the responses (returned by environment) that are useful to characterize the the key elements contained in the underlying image.

**Overall Setup.** Though this environment does not involve actually HF, the policies' performance intrinsically pertain to two types of returns as per the setup from [66]. Specifically, the immediate rewards, used to train the language generation policy (captured by generative pre-trained transformer version 2, or GPT2) to asks questions, are discrete, *i.e.*, a $-1$ reward is given at each step until most key elements can be inferred from the dialogue, and an additional $-1$ reward is given if the questions asked by the agent lead to responses that only contain trivial information towards extrapolating the image characteristics (such as the responses that only contain a 'yes/no'). In contrast, a separate pre-trained referee model is used to determine when sufficient elements in the underlying images can be captured from the dialogue history and terminates the episode, which maps the dialogue history to embeddings that represent the features pertaining to the image elements that can be extrapolated from the dialogue, followed by scoring the dialogue with relative percentile ranking of the ground truth image's distance from the predicted embedding among a set of images taken from the evaluation set provided by [66, 10], *i.e.*, an episode is terminated if the percentile ranking is improved by at least 50% compared to the percentile ranking before the dialogue begins, $(1 - p_T) \geq .5 \cdot (1 - p_0)$, where $p_T$ is the ground truth image's percentile rank at dialogue's last turn and $p_0$ is the ground truth image's percentile rank at the beginning of the dialogue, when only the image caption is observed. As

Table 10: Environmental and human returns of all the target policies pertaining to **Patient #2**, from the adaptive neurostimulation experiment.

| Target Policy | Environmental Returns | Human Returns (x10) |
|:---:|:---:|:---:|
| #0 | -142.20 | 160.42 |
| #1 | -79.18 | 163.55 |
| #2 | -155.11 | 158.53 |
| #3 | -152.65 | 165.62 |
| #4 | -154.66 | 161.03 |
| #5 | -80.63 | 69.42 |

Table 11: Environmental and human returns of all the target policies pertaining to **Patient #3**, from the adaptive neurostimulation experiment.

| Target Policy | Environmental Returns | Human Returns (x10) |
|:---:|:---:|:---:|
| #0 | -60.78 | 154.89 |
| #1 | -83.83 | 66.20 |
| #2 | -126.97 | 142.45 |
| #3 | -154.83 | 129.57 |
| #4 | -147.44 | 142.29 |
| #5 | -165.74 | 143.39 |

a result, the former type of rewards are considered as the *environmental rewards* as per the OPEHF setup, and the latter one is closely related to the human returns as it is only provided at the end of each episode and are obtained following a different schema. Specifically, we consider $p_0 - p_T$ as the *synthetic* human return, which quantifies the improvement over the percentile ranking throughout the dialogue. More details on the environmental reward function and the percentile ranking generated by the referee model can be found in [66].

**MDP Formulation.**    Consequently, in this experiment we compare OPEHF's performance against baselines, toward predicting the percentile rankings corresponding to the embeddings predicted from the referee model at the end of episodes. The MDP states are captured by the embeddings output by the attention layer from the encoder of the GPT2, by feeding in the dialogue history up to the current step. The action space is captured as $\mathcal{V}^l$, where $\mathcal{V}$ is the vocabulary considered and $l$ is the maximum number of words the agent is allowed to place in the questions in each step. The definition of environmental rewards and synthetic human returns are introduced above.

**Offline Trajectories and Target Policies.**    Five behavioral policies are used to collect offline trajectories, which are obtained from training over 4 types of RL algorithms following the setup introduced in [66], *i.e.*, 1 policy follows conservative Q-learning (CQL), 2 follow implicit Q-learning (IQL), 1 follows behavioral clone (BC) and the last follows decision transformer (DT). All the behavioral policies are trained to obtain only 10%-50% of the best synthetic human returns that can be achieved, which are considered sub-optimal. Eight different policies (also obtained from 4 types of RL algorithms) constitute the set of target policies whose performance will be estimated by OPEHF and the baselines. Their performance spread roughly equally between 0%-100% of the optimal return that could be achieved; details can be found in Table 13. It can be found that synthetic human returns can be negative if the dialogue provides trivial, or sometimes misleading, information that are not helpful for capturing the underlying image. In practice one can shift the returns to become all positive in order to comply with the assumption in Proposition 1. Moderate correlations are found between the environmental and synthetic human returns in this experiment, *i.e.*, Pearson's correlation coefficient is 0.606 and Spearman's rank correlation coefficient is 0.69.

**Results and Discussion.**    The results are summarized in Table 12. It can be observed that the OPEHF in general outperforms the baselines, with significantly lower MAEs and higher ranks if IS, DR or FQE are selected as the downstream estimator. We also find that if DICE is selected as the downstream estimator, it in general leads to lower MAEs and regrets for both the OPEHF and rescale baseline, but significantly low rank correlations. Our hypothesize would be that considering DICE directly

Table 12: Results from the visual Q&A environment.

| | IS | | | DR | | |
|---|---|---|---|---|---|---|
| | Fusion | Rescale | **OPEHF** (our) | Fusion | Rescale | **OPEHF** (our) |
| MAE | 1.13±0.41 | 0.40±0.27 | **0.56±0.12** | 1.27±0.33 | 0.72±0.21 | **0.63±0.08** |
| Rank | -0.21±0.55 | 0.19±0.15 | **0.64±0.13** | 0.11±0.45 | 0.08±0.27 | **0.59±0.18** |
| Regret@1 | 0.76±0.34 | **0.08±0.09** | 0±0.05 | 0.43±0.29 | 0.25±0.09 | **0.01±0.0** |

| | DICE | | | FQE | | |
|---|---|---|---|---|---|---|
| | Fusion | Rescale | **OPEHF** (our) | Fusion | Rescale | **OPEHF** (our) |
| MAE | 0.99±0.12 | 0.25±0.03 | **0.11±0.02** | 0.97±0.08 | 1.13±0.14 | **0.89±0.03** |
| Rank | **0.02±0.24** | -0.13±0.1 | **0.06±0.32** | 0.1±0.05 | 0.12±0.52 | **0.55±0.28** |
| Regret@1 | **0.01±0.01** | 0.07±0.0 | **0.02±0.03** | **0.01±0.0** | 0.03±0.03 | **0.01±0.0** |

Table 13: Environmental and synthetic human returns of the target policies, as well as the algorithms used to obtain them, considered in the Q&A dialogue experiment.

| Target Policy | Environmental Returns | Synthetic Human Returns | Algorithm |
|---|---|---|---|
| #0 | -15.46 | -18.15 | CQL |
| #1 | -11.77 | -3.20 | IQL |
| #2 | -10.86 | 1.09 | CQL |
| #3 | -13.39 | 2.24 | CQL |
| #4 | -13.50 | 2.36 | BC |
| #5 | -11.33 | 3.09 | CQL |
| #6 | -6.05 | 3.23 | IQL |
| #7 | -9.77 | 3.66 | DT |

estimate the discrepancy between target and behavioral policy's propensity of visiting particular state-action pairs across the offline trajectories, it may benefit from leveraging the information (over the state-action space) captured by a language encoder that place roughly equal attention to each step of the dialogue, as opposed to the GPT2's encoder which is not fine-tuned specifically for such a case.

