# OpenReview forum: "Off-Policy Evaluation for Human Feedback"
_NeurIPS.cc/2023/Conference — NeurIPS 2023 poster_

### Official Review · Reviewer_cuXN · 2023-06-28

**Soundness:** 3 good
**Presentation:** 3 good
**Contribution:** 3 good
**Rating:** 6
**Confidence:** 3

**Summary:**

This paper proposes an off-policy evaluation method that utilizes sparse human feedback. Experiments are conducted on two real-world environments and one simulation environment.

**Strengths:**

The idea of incorporating human feedback into policy evaluation is interesting and significant. The proposed framework is compatible with existing methods. The experiments are comprehensive. Especially, real-world experiments are appreciated.

**Weaknesses:**

My major concerns are described in the Questions section. It would be appreciated if the authors address them in detail and I’m happy to raise my score.

**Questions:**

1.	Are 4 patients enough for validating the method? I’m wondering how many participants other similar methods adopt.
2.	The proposed method additionally requires offline human feedback data. Are these comparing baselines also using the extra human data? If not, what did you do to make fair comparisons?


**Limitations:**

Please see the questions.

---

> ### Author Rebuttal · Authors · 2023-08-09
>
> We greatly appreciate the reviewer's thoughtful questions and constructive feedback. Please find our responses below. We hope they addressed the reviewer's outstanding concerns sufficiently, and we would be more than happy to further respond to any follow-ups the reviewer may have.
>
> Q1: Are 4 patients enough for validating the method? I’m wondering how many participants other similar methods adopt.
>
> R1: Good question. The scale of our cohort is limited in the sense that a special programmable DBS pulse generator is needed to be implanted to the patient's chest (see Figure 6 in Appendix), in order to deploy RL policies that can adapt stimulation amplitudes in real-time. As a result, it was challenging to find patients who are willing to participate in the study, starting with implanting a non-ordinary device followed by experimenting with relatively new control strategies in DBS. In contrast, most existing devices only simply support turning on/off stimulations, and some are not even adjustable in any cases [1, 2]. Under the topic of amplitude-adaptive DBS, we had found one existing work that tested a proportional-integral controller over 5 patients [3], and another one that tested over 13 patients but only 8 hours per patient [4]. On the other hand, though a limited number of patients were available, each one had been followed up for at least 2 years, leading to nearly hundred-hour of logged data for OPEHF to be used for training and validation. Consequently, the performance comparison results are considered statistically significant over each patient. Please refer to our response to reviewer ochq's Q4, on how we plan to dedicate our future efforts on creating more robust and significant evaluation procedures along the line of OPEHF.
>
> Q2: The proposed method additionally requires offline human feedback data. Are these comparing baselines also using the extra human data? If not, what did you do to make fair comparisons?
>
> R2: Yes. The baseline called `rescale` directly assigned the offline human feedback received at the end of each episode as the reward for the last step, and divided it by $\gamma^T$ ($T$ is the trajectory horizon) to ensure that the discounted sum of rewards equals the feedback received. The other baseline called `fusion` additionally includes the immediate environmental rewards at each step, and then add in the human feedback to the reward at the last step, still with a correction applied such that the cumulative sum of discounter rewards in each trajectory is equal to the human feedback provided. We discussed this in between lines 226 and 236, and we would be more than happy to clarify any parts that are unclear there in the camera ready, if our paper is accepted. We thank the reviewer for paying attention to details.
>
> [1] Beudel, M., and P. Brown. "Adaptive deep brain stimulation in Parkinson's disease." Parkinsonism & related disorders 22 (2016): S123-S126.
>
> [2] Little, Simon, et al. "Adaptive deep brain stimulation for Parkinson's disease demonstrates reduced speech side effects compared to conventional stimulation in the acute setting." Journal of Neurology, Neurosurgery & Psychiatry 87.12 (2016): 1388-1389.
>
> [3] Malekmohammadi M, Herron J, Velisar A, et al. Kinematic adaptive deep brain stimulation for resting tremor in Parkinson's disease. Mov Disord 2016;31:426-428
>
> [4] Arlotti M, Marceglia S, Foffani G, et al. Eight-hours adaptive deep brain stimulation in patients with Parkinson disease. Neurology 2018;90:e971-e976

---

> > ### Comment · Reviewer_cuXN · 2023-08-14
> > **Response to the authors**
> >
> > Thanks for the authors' detailed responses. I'm happy to raise my score to 6.

---

> > > ### Author Response · Authors · 2023-08-16
> > >
> > > Thank you for acknowledging our responses. We greatly appreciate your time and efforts toward getting our work to be thoroughly evaluated.

---

### Official Review · Reviewer_1oyy · 2023-07-07

**Soundness:** 2 fair
**Presentation:** 3 good
**Contribution:** 2 fair
**Rating:** 5
**Confidence:** 3

**Summary:**

The authors examine an off-policy evaluation scenario where we have an environment with an intrinsic environment reward $r$, and an extrinsic human reward $r^H$. A further assumption is made that the human reward is only observed at the end of the episode, and comes as an rating on the total return of the episode. This is done to line up with cases where human ratings may only be clear after the fact.

The goal of the paper is to do off-policy evaluation in this modified class of MDPs. Two baseline approaches for doing this are considered. The first is to treat the human return as a reward that is achieved at the last timestep of the episode, with $r = 0$ at all earlier steps. The second is to set the human reward $r^H = r$, where $r$ is the original known environment reward, then add an offset $G^H - G$ term to the last timestep, where $G^H$ = total human return and $G$ = total environment return.

This paper proposes a method to learn timestep level $r^H$ directly. First it argues that given accurate timestep level $r^H$, the variance of the policy return estimate is equal or smaller than the estimate in the sparse MDP where all human reward is allocated to the final timestep. The reward learning method is then done based on two objective terms. The total reward must equal the human label of return for each trajectory, and the reward for individual $(s,a)$ should be similar to reward for other $(s',a')$ that have similar embeddings.

This then leaves determining the embeddings. This is done via a VAE to learn a latent space, and the the nearest K-neighbors of a given $(s,a)$ embed are treated as similar examples.

Once a reward estimate is learned, any off the shelf OPE method can be used.

**Strengths:**

The paper makes an important observation that human feedback tends to be given at the end of an episode or after the fact, and is usually not given at the per-timestep level. It is clear that if per-timestep labels can be learned, then using them to estimate policy performance should give better results than estimating just total episode return (at least assuming learning per-timestep labels is a well-formed problem)

**Weaknesses:**

I don't like that the paper proposes an "OPEHF framework", yet the bulk of the paper is really about learning a reward function / learning to distribute a return estimate over past trajectories. Nothing about the reward function learning is inherently tied to off-policy evaluation, it could theoretically be done online too, it's just something that can be done off-policy as well.

The idea of return decomposition in the paper seems very similar to the RUDDER: Return Decomposition for Delayed Rewards paper, albeit with a different use case (in that paper, the environment naturally has a large sparse reward, which is decomposed to provide partial reward signal for an RL agent instead of needing to learn to reachieve the large sparse reward seen before.) It seems very relevant, in that the RUDDER paper proposes a number of return decomposition methods, none of which are considered in this paper.

The embedding of $(s,a)$ is said to come from a bi-directional LSTM, in order to leverage information from both previous and subsequent steps. Doesn't this break the Markov assumption of MDPs? Since the rewards $r(s,a)$ now may depend on future states?

How does the method handle cases where $r(s,a)$ may not be well defined? For example, two human raters have differing opinions on the score of a given trajectory.

Overall I do not feel the paper is benchmarked sufficiently to be accepted.

Edit: see rebuttal chain, have decided to increase score based on discussion of concerns.

**Questions:**

Are there examples of ways the original environment reward can be used, aside from the fusion baseline mentioned? I am trying to understand if it makes sense to even view this as a new class of MDPs, vs treating it like an MDP with one reward function that is only observed at the last timestep. It seems like the method proposed here does not use original env reward in any way besides training the encoder-decoder in the VAE?

**Limitations:**

Yes

---

> ### Author Rebuttal · Authors · 2023-08-09
>
> Please find our responses below. We hope they addressed the reviewer's outstanding concerns sufficiently, and we would be more than happy to respond to any follow-ups the reviewer may have.
>
> Q1:The bulk of the paper is really about learning a reward function. It could theoretically be done online too.
>
> R1: We respectfully disagree that our work is more of a reward shaping work, nor can be adapted to the online learning setup, from two perspectives.
>
> * The motivation for decomposing human returns in our case, follows from the analysis in proposition 1 that proper decomposition can help reduce variance and lead to **more accurate OPE estimation over policies that lead to varied performance**. This is rather different than in existing reward shaping works that try to smooth over sparse rewards, and mainly *focus on improving the convergence and/or optimality properties of the optimal policy* that can be learned out of the shaped reward functions.
>
> * The focus of our work is to estimate the feedback/returns that are directly provided by human participants, toward the cases where the performance pre-existing RL policies need to be demonstrated **before they get deployed to human participants**, e.g., as required in healthcare systems. So the problem setup of OPEHF cannot be changed to online. Moreover, the use of bi-directional LSTM requires all trajectories to be fully observed *a priori*, and cannot be used online as well (see more in R3 below).
>
>
> Q2: The idea of return decomposition in the paper seems very similar to the RUDDER paper, albeit with a different use case.
>
> R2: We thank the reviewer for sharing this paper. After reading it, we found that RUDDER and our work have different targets, and it would not be very straightforward to adapt the setup considered in one into another. Specifically, for two reasons, i.e.,
>
> * RUDDER is a policy optimization work that focuses on designing the rewards such that the high-return state-action pairs can be quickly identified and frequently re-visited, by allowing the value function to follow shapes like step functions. RUDDER's framework only guarantees that the optimal policy would not change under the MDP before and after reward shaping. However, it does not discuss/prove the rank of returns over sub-optimal policies will be preserved, where in OPE it is important that the returns of policies that lead to varied performance can be estimated accurately [1, 2].
>
> * It seems that RUDDER requires online updates where new data is constantly being added to the buffer during training, which does not align with the OPE where the agent only has access to a **fixed** set of **offline trajectories** (no new data can be added).
>
> Nevertheless, it was a very interesting read and we can cite this work and discuss how the ideas introduced in RUDDER could potentially facilitate future works down the line of OPEHF, if our paper is accepted.
>
> Q3: The embedding of $(s,a)$ is said to come from a bi-directional LSTM. Doesn't this break the Markov assumption of MDPs?
>
> R3: The bi-directional LSTM is used to map the end-of-episode human return back to immediate human rewards (IHRs) over the entire horizon, for all the *offline trajectories* being considered. Specifically, **it is used only after these trajectories are already observed** (i.e., no online deployment nor adding new trajectories). Such a case falls under the typical sequence modeling setup where RNNs/LSTMs are mainly used. As an analogy, the use case here is somewhat reminiscent of how decision transformers [3, 4] are used to model the decision process.
>
> Q4: How does the method handle cases where $r(s,a)$ may not be well defined? For example, two human raters have differing opinions on the score of a given trajectory.
>
> R4: In this work we considered that each trajectory only receives a single human return **provided by the participants themselves** at the last step (see section 2.1). For example, pertaining to the physiological and/or subjective feedback in our DBS experiments. So there would not be conflict cases. Please refer to our response to reviewer ochq's Q4 for why we could not follow RLHF-styled evaluations (by hiring 3-rd party labellers). Similar setups are adopted in other OPE works [5, 6], and are considered more challenging than evaluating labeller's ratings, since human returns/feedback can be intrinsically affected by multiple underlying factors that are not very easy to be captured.
>
> Q5: Overall I do not feel the paper is benchmarked sufficiently to be accepted.
>
> R5: To evaluate our method one needs access to records that both log the trajectory and participant ratings, where we agree that currently it lacks a standard testing framework for this. However, both real-world experiments we presented are backed with substantial data collected from extensive testing (over years), where results demonstrated statistical significance. Please also see our general response above, as well as response to reviewer ochq's Q1 for details on the **additional natural language Q&A experiment we had in Appendix E**.
>
> Q6: Are there examples of ways the original environment reward can be used, aside from the fusion baseline mentioned? It seems like the method proposed here does not use original env reward in any way besides training the encoder-decoder in the VAE?
>
> R6: The experiments (Tables 1 and 3) have shown that the immediate environmental rewards may not correlate very well with the human returns (i.e., mostly weak-to-moderate correlations), due to the nature of human feedback [5, 6]. So in our methodology the environmental rewards have only been used as part of supplementary information to train the VLM-H to better capture the underlying dynamics. We would leave for future works to explore how to further leverage the environmental rewards. We appreciate this thoughtful question that expands the future avenues following our work.

---

> > ### Author Response · Authors · 2023-08-10
> > **References**
> >
> > [1] Fu, Justin, et al. "Benchmarks for Deep Off-Policy Evaluation." International Conference on Learning Representations. 2020.
> >
> > [2] Zhang, Michael R., et al. "Autoregressive Dynamics Models for Offline Policy Evaluation and Optimization." International Conference on Learning Representations. 2020.
> >
> > [3] Chen, Lili, et al. "Decision transformer: Reinforcement learning via sequence modeling." Advances in neural information processing systems 34 (2021): 15084-15097.
> >
> > [4] Janner, Michael, Qiyang Li, and Sergey Levine. "Offline reinforcement learning as one big sequence modeling problem." Advances in neural information processing systems 34 (2021): 1273-1286.
> >
> > [5] Namkoong, Hongseok, et al. "Off-policy policy evaluation for sequential decisions under unobserved confounding." Advances in Neural Information Processing Systems 33 (2020): 18819-18831.
> >
> > [6] Chesnaye, Nicholas C., et al. "An introduction to inverse probability of treatment weighting in observational research." Clinical Kidney Journal 15.1 (2022): 14-20.

---

> > ### Comment · Reviewer_1oyy · 2023-08-17
> >
> > Apologies for delay, was addressing another deadline.
> >
> > > R1: We respectfully disagree that our work is more of a reward shaping work, nor can be adapted to the online learning setup, from two perspectives.
> > > The motivation for decomposing human returns in our case, follows from the analysis in proposition 1 that proper decomposition can help reduce variance and lead to more accurate OPE estimation over policies that lead to varied performance. This is rather different than in existing reward shaping works that try to smooth over sparse rewards, and mainly focus on improving the convergence and/or optimality properties of the optimal policy that can be learned out of the shaped reward functions.
> > > The focus of our work is to estimate the feedback/returns that are directly provided by human participants, toward the cases where the performance pre-existing RL policies need to be demonstrated before they get deployed to human participants, e.g., as required in healthcare systems. So the problem setup of OPEHF cannot be changed to online. Moreover, the use of bi-directional LSTM requires all trajectories to be fully observed a priori, and cannot be used online as well (see more in R3 below).
> >
> > I would argue that generally, improving the convergence the policy via reward shaping is *because* reward shaping reduces the variance of the policy update. Perhaps the more accurate way to put it is that both OPEHF and general reward shaping are trying to reduce variance, but one is for evaluation and the other is for improving gradient updates.
> >
> > I think that overall this is not an important concern though and am willing to move on.
> >
> > > On RUDDER
> >
> > The authors are correct that RUDDER's return decomposition view only ensures the optimal policy's return is unchanged. I was more curious at the different variants RUDDER proposes for return decomposition (see Section 3 of paper: integrated gradients, layer-wise relevance propagation, etc.)
> >
> > Could the authors clarify if OPEHF will maintain the ranks of different policies in the limit of infinite data? It was not clear to me that was a proposed contribution.
> >
> > > R3: The bi-directional LSTM is used to map the end-of-episode human return back to immediate human rewards (IHRs) over the entire horizon, for all the offline trajectories being considered. Specifically, it is used only after these trajectories are already observed (i.e., no online deployment nor adding new trajectories). Such a case falls under the typical sequence modeling setup where RNNs/LSTMs are mainly used. As an analogy, the use case here is somewhat reminiscent of how decision transformers [3, 4] are used to model the decision process.
> >
> > I think decision transformers are not a great analogy here. In decision transformers, we condition on the return to go, but this does not change the reward function $r(s,a)$ of the underlying MDP. That reward function is still Markovian / does not depend on the future. Although at training time DT is aware of the future (knows the true return to go), at inference time it is not aware of the future and we pick a return-to-go to condition on at $t=0$.
> >
> > In OPEHF, the reward $r(s,a)$ does depend on the future, via the bi-directional part of the LSTM. And, this is all fine for off-policy evaluation where we do not do online execution, but I would argue it's not exactly a reward function anymore. You would literally be unable to use the $r(s,a)$ during live inference if I understand correctly.
> >
> > To be clear: for the purposes of OPE, the bi-directional LSTM is probably helpful. Since the human giving the preference label sees the entire trajectory $\tau$ before rating the trajectory. I just would not describe it as $r(s,a)$ when it is more like $r(s,a|\tau)$ in practice.
> >
> > > Challenges in getting more than 1 rater for a scenario.
> >
> > I will quote the reply to Q4 from reviewer ochq.
> >
> > > Unlike frameworks like RLHF for language model training [5], where one can hire 3rd-party raters to provide feedback simply by following a predefined rubric (without the need of domain expertise); in human-centric domains like healthcare or education, professionals that are thoroughly trained (e.g., doctors, professors, instructors) over years may only be eligible to rate, other than the participants themselves.
> >
> > I agree with all of this, but think of the question as, *in a hypothetical scenario where there is more than 1 human rater, what would happen?* I am not expecting experiments for this, I understand the setting in this paper has 1 human rater and more would be difficult. I am more looking for discussion on if the method could handle multiple $r(s,a)$ labels gracefully or not. If it cannot it would be a good limitation to mention.
> >
> > > Evaluation
> >
> > To be specific, I was interested in comparisons to other methods of return decomposition. i.e.  the proposal of this work is, "reconstruct IHR, here is 1 way to do it" and I wanted to see > 1 way of doing it (from the existing data, to avoid needing to get more)

---

> > > ### Author Response · Authors · 2023-08-18
> > > **Author's responses**
> > >
> > > We thank the reviewer for getting back to us in time. Please see our detailed responses below.
> > >
> > > Q7: I would argue that generally, improving the convergence the policy via reward shaping is because reward shaping reduces the variance of the policy update ...... **I think that overall this is not an important concern though and am willing to move on.**
> > >
> > > R7: Thank you for aligning with us that our work does not simply fall within the scope of reward shaping, echoing on our R1 above.
> > >
> > > Q8: **The authors are correct that RUDDER's return decomposition view only ensures the optimal policy's return is unchanged.** I was more curious at the different variants RUDDER proposes for return decomposition (see Section 3 of paper: integrated gradients, layer-wise relevance propagation, etc.) Could the authors clarify if OPEHF will maintain the ranks of different policies in the limit of infinite data? It was not clear to me that was a proposed contribution.
> > >
> > > R8: Reward decomposition is part of RUDDER’s methodology, with the final goal as constructing a **transformed MDP that has expected future rewards equal to zero**, from which Q-values can be captured smoothly. This is helpful for policy optimization as the agent mainly need to act following argmax of leanred Q values there to find the optimal policy, hence RUDDER only guarantees that the optimal policy under the pre- and post-transformation MDPs are equivalent.
> > >
> > > However, **it dose not guarantee that, under the transformed MDP, the redistributed rewards can properly facilitate downstream OPE estimators**. As this would require one to show that not only the optimal policy between two MDPs are equivalent, but all the sub-optimal policies can result in the same return distribution before and after transformation; because policies with varied scale of returns will be evaluated by OPE [1,2 above], not only just the optimal ones. **In contrast, our framework did not rely on any MDP transformations, thus would not bounded by this limitation.**
> > >
> > > To this end, we believe that it would require non-trivial efforts to integrate the return decomposition method in RUDDER into the OPEHF framework. However, as we mentioned, we appreciate the reviewer sharing an interesting read, and we would be more than happy to discuss in the camera-ready how similar ideas in RUDDER can be adapted toward OPEHF in the future, if accepted. We believe one of the (future) adaptation could be using this idea to further improve FQE estimators, as part of the downstream OPE estimator we considered.
> > >
> > > Q9: In OPEHF, the reward $r(s,a)$ does depend on the future, via the bi-directional part of the LSTM. And, this is all fine for off-policy evaluation where we do not do online execution, but I would argue it's not exactly a reward function anymore. You would literally be unable to use the $r(s,a)$ during live inference if I understand correctly. **To be clear: for the purposes of OPE, the bi-directional LSTM is probably helpful.** Since the human giving the preference label sees the entire trajectory before rating the trajectory. I just would not describe it as $r(s,a)$  when it is more like $r(s,a|\tau)$  in practice.
> > >
> > > R9: Thank you for alingning with us on the purpose of using bi-directional LSTM for RILR. We wanted to emphasize that the immediate human rewards $r^\mathcal{H}(s_t,a_t)$ follows Markov property, as defined in Section 2.1. However, with $r^\mathcal{H}(s_t,a_t)$ being unknown, the RILR uses bi-directional LSTM to approximate $\hat r^\mathcal{H}(s_t,a_t)$ which may depend on future states. We agree with the reviewer that we can correct the notation $\hat r^\mathcal{H}(s_t,a_t)$ to be $\hat r^\mathcal{H}(s_t,a_t|\tau)$ in the camera-ready. Since we are considering OPE where we only work with a fixed offline dataset, we would not encounter the cases that require ‘live inference’.
> > >
> > > Q10: **I agree with all of this**, but think of the question as, in a hypothetical scenario where there is more than 1 human rater, what would happen?
> > >
> > > R10: Our work specifically consider the case where the participants directly provide feedback, as defined in Section 2.1. It is mentioned as part of our future work where we are building a platform to allow professionals to rate the trajectories as well, and there definitely need some extra work to adapt the current OPEHF frameowrk for that, as it is beyond the scope of this paper.

---

> > > > ### Author Response · Authors · 2023-08-18
> > > > **Author's response (cont.)**
> > > >
> > > > Q11: To be specific, I was interested in comparisons to other methods of return decomposition. i.e. the proposal of this work is, "reconstruct IHR, here is 1 way to do it" and I wanted to see > 1 way of doing it (from the existing data, to avoid needing to get more)
> > > >
> > > > R11: Besides the discussion in R8, we wanted to further emphasize that most existing reward decomposition work **decompose the delayed environmental returns**, leveraging the fact that some state-action pairs may contribute more to the final returns, following typical environmental dynamics in, for example, games and robotics. However, the human returns in OPEHF is de-coupled from the environmental returns, which is justified in experiments that low correlations are found between them in general. Multiple works have found that it does not simply follow the way how environmental returns would be composed, as it depends on many factors even outside of scope of what the environmental dynamics could capture, such as the mood when a participant provides feedback and more other underlying confounders that are non-trivial to capture [5,6 above]. We believe that our current work is both technically and experimentally complete, and solves an important empirical problem within the scope of OPE (as pointed out by reviewers KHGy, ochq and cuXN). As we are, to the best of our knowledge, the first work that introduce the framework for OPEHF, we tried to make our approach straightforward, to maximize the comaptibillity with existing OPE frameworks. Some components in our framework could potentially be further improved by leveraging the ideas from related communities, and we appreciate the reviewer pointing to us one possible future avenue.
> > > >
> > > >
> > > > Again, we greatly appreciate the reviewer’s efforts facilitating meaningful discussions with us, and we are delighted to see that the reviewer has aligned with us on many of the questions above. We are always more than happy, and grateful, to have the chance addressing any further comments. We wish the reviewer all the best for the endeavor catching up the other deadline.

---

> > > > > ### Author Response · Authors · 2023-08-18
> > > > > **Any follow-ups?**
> > > > >
> > > > > As the discussion window is closing in the next 2 days. We are wondering if the reviewer would have any follow-ups over our latest response. We would also greatly appreciate the reviewer to **reconsider the assessment and recommendation** of our work, as we have seen that the reviewer had agreed with us on many points above (as highlighted when we quote the reviewer's responses).

---

> > > > > > ### Comment · Reviewer_1oyy · 2023-08-19
> > > > > >
> > > > > > > R7: Thank you for aligning with us that our work does not simply fall within the scope of reward shaping, echoing on our R1 above.
> > > > > >
> > > > > > To be clear: I still pretty strongly *disagree* on this point. I just think it is not relevant to discussion on the quality of the paper. Please don't imply that I agree here.
> > > > > >
> > > > > > > R8: On RUDDER and OPEHF:
> > > > > >
> > > > > > I don't think the transformed MDP in RUDDER is particularly important. The transformed MDP accumulates reward-so-far into an extra feature just to keep the Markovian assumption true. But if done in an OPE way, then we do not need to keep that Markovian assumption, right? That is what the authors are arguing. RUDDER does not guarantee the transformed reward facilitates OPE - but then, neither does the author's method. Both are about how the VAE-based embedding works in practice.
> > > > > >
> > > > > > > R9: On Markovian reward:
> > > > > >
> > > > > > If the authors mention the distinction here that the approximate $\hat{r}$ is not a true reward function, is conditional on $\tau$, and is not usable for "live inference", then I am satisfied on this point.
> > > > > >
> > > > > > (I do think there is an interesting question here about if $r(s,a)$ is even a suitable model for human reward - see http://ai.stanford.edu/blog/robomimic/ for an argument that human decision making is often not Markovian due to unobserved state. Might be of interest to authors.)
> > > > > >
> > > > > > > R10: More than 1 human rater:
> > > > > >
> > > > > > Please include this as a limitation of the work as well.
> > > > > >
> > > > > > > R11: Other return decomposing methods.
> > > > > >
> > > > > > Here, the authors say that
> > > > > > > However, the human returns in OPEHF is de-coupled from the environmental returns, which is justified in experiments that low correlations are found between them in general. Multiple works have found that it does not simply follow the way how environmental returns would be composed, as it depends on many factors even outside of scope of what the environmental dynamics could capture, such as the mood when a participant provides feedback and more other underlying confounders that are non-trivial to capture [5,6 above]
> > > > > >
> > > > > > If human returns are decoupled from environmental returns because of factors outside the scope of environmental dynamics, how is that handled by the proposed methods in OPEHF? The OPEHF embeddings are based solely on fitting from the environmental dynamics sequence $(s,a,s,a,\cdots)$ to the human return label $R$.
> > > > > >
> > > > > > I think at a theory level it's all equally intractable, but then deep learning finds a way. Which is to say that, this argument that other return decomposing methods are less relevant does not feel strong to me.
> > > > > >
> > > > > > -----------------------------
> > > > > >
> > > > > > I thank the authors for replying to my comments so far. I'd like to let them know that I'll be busy tomorrow and won't be available to respond (but they are welcome to write a reply for other reviewers to read if they wish). I plan to bump my 3 to a 5, I still have a number of reservations about the work but am feeling more borderline about it than the 1st pass.

---

> > > > > > > ### Author Response · Authors · 2023-08-20
> > > > > > > **Thank you**
> > > > > > >
> > > > > > > We thank the reviewer for sharing these thoughtful comments, and we indeed appreciate the feedback/thoughts from different aspects. We will definitely reflect them in our camera ready version, if accepted.

---

### Official Review · Reviewer_KHGy · 2023-07-07

**Soundness:** 3 good
**Presentation:** 3 good
**Contribution:** 2 fair
**Rating:** 6
**Confidence:** 3

**Summary:**

This paper discusses a framework that learns to decompose human feedback reward, which is usually sparse and only given at the end of the episode, to intermediate step reward. They formulate a reward decomposition objective that considers original environment trajectory information using a variational auto-encoding representation learning objective as an implicit constraint. The authors show that their framework can reduce MAE for multiple OPE algorithms on a few domains.

The core contribution of this paper is composed of a few known techniques, but the final result is an enhanced OPE with a lower MAE. The real-world experiments are the highlight of this paper.

**Strengths:**

The paper has offered two very interesting real-world experiments to demonstrate the effectiveness of the proposed framework and show how it can enhance existing OPE algorithms. Indeed, a lot of existing OPE algorithms often work well with dense rewards, and these algorithms do not have an additional component that can infer intermediate rewards.

I find the real-world experiments very refreshing and quite convincing.

**Weaknesses:**

The methodology proposed in this paper is not very novel — the authors assembled a few known techniques in the AI/ML community. Also, I’m not entirely sure some of the content in the paper is necessary. The authors can correct me if I’m wrong or misunderstood the contribution, but some parts of the paper do not seem like original contributions but simply restatements of known results.

Line 101: Problem 1. The authors indicated one of the issues is “the total number of offline trajectories AND $r_t^H$ being unknown.” However, in the later sections, I think only $r_t^H$ was solved. It doesn’t seem like the proposed framework would change much if “the total number of offline trajectories” is known or unknown. Can authors provide more insight on how this affects their framework design? Specifically, how would they change the algorithm if the number of trajectories is known? I sincerely suggest dropping this if this is not a serious factor that affects your algorithm design.

Line 121 Proposition 1: The rescaled OPE estimator has a lower variance than the original OPE estimator. I think this is a known result. For example, weighted importance sampling has a lower variance than importance sampling. Similarly, weighted PDIS has a lower variance than PDIS. See Thomas [1] Chapter 3 for consistent weighted PDIS. Can the authors explain and clarify how they made a novel contribution beyond what’s provided in [1]? Otherwise, I suggest the authors cite the previous work so readers will not get confused.

[1] Thomas, P. S. Safe Reinforcement Learning. PhD thesis, University of Massachusetts Amherst, 2015b.

**Questions:**

1. FQE is an OPE algorithm that tries to do reward decomposition. However, the experiment still showed that using the decomposed reward produced from OPEHF, FQE-OPEHF can have a lower MAE than just using FQE alone. Can the authors comment on why this would be the case?
2. The objective is on the sum of discounted rewards being the same, but there is no constraint on reward from each timestep — this means “theoretically” we can have a neural network predict -10000 on one timestep and 9999 on the next step, and still have the sum of reward to be -1. Can the authors comment on why this degenerate case hasn’t happened, and can authors provide some analysis/insight/summary data on what the decomposed intermediate human feedback rewards look like?

**Limitations:**

The authors addressed the limitations of their approach.

---

> ### Author Rebuttal · Authors · 2023-08-09
>
> We greatly appreciate the reviewer's thoughtful questions and constructive feedback. Please find our responses below. We would be more than happy to further respond to any follow-ups the reviewer may have.
>
> Q1: The methodology proposed in this paper is not very novel — the authors assembled a few known techniques in the AI/ML community.
>
> R1: To the best of our knowledge, our work is the first to introduce a practical framework that allows OPE to accurately estimate human returns that are only provided at the end of each episode, which has been a long-standing challenge in OPE as the reviewer pointed out in the Strengths section of the review. Our methodology is designed to be straightforward, allowing the maximum compatibility with existing down-stream OPE methods, without reinventing the wheels.
>
> The main **technical novelty** lies in formulating the surrogate objective for capturing the immediate human rewards (IHRs) in eq.(1), and solving it by regularizing the reconstructed rewards over state-action pairs that achieved similar embeddings in the latent space (eq.(3) and (4)). The VLM-H and bi-directional LSTM models were used to capture environmental dynamics and reconstructing IHRs respectively, however, the bi-directional LSTM leverages the latent information encoded by VLM-H to map end-of-episode human returns to IHRs. Consequently, each component is introduced toward a specific sub-goal, and they are all closely tied with each other in order to achieve the overall objective.
>
> Moreover, we have shared an **important empirical finding** that could be useful for future works to leverage, when shaping the problem setup along the line of OPEHF. Since only weak-to-moderate correlations are found between human returns and environmental rewards (as a result of the two real-world experiments), two separate reward functions would be needed when formulating the MDP setup.
>
> Q2: Line 101: Problem 1. The authors indicated one of the issues is “the total number of offline trajectories AND $r_t^\mathcal{H}$ being unknown.”
>
> R2:  We thank the reviewer for pointing this out. We meant to only consider the IHRs, $r_t^\mathcal{H}$, to be unknown. This sentence is rather long and the writeup there may cause this confusion here -- we will correct it if the paper gets accepted.
>
> Q3: Line 121 Proposition 1: The rescaled OPE estimator has a lower variance than the original OPE estimator. I think this is a known result.
>
> R3: We thank the reviewer for sharing this relevant literature and we will cite it in the camera-ready, if our paper is accepted. Proposition 1 is used to illustrate the motivation why in our methodology it needs to map the end-of-episode human returns to each step. The discussion there is specific to our problem setup (section 2.1) under the human reward function considered within the MDP, where $r_t^\mathcal{H}$ is considered a random variable sampled from a distribution. This is due to the fact that human rewards can be affected by multiple underlying factors [1,2], which could be non-deterministic even if the $(s_t,a_t)$ pair is already observed. The analysis then follows from here, whereas in the literature provided by the reviewer, it mainly assumes that rewards are scalar once $(s_t,a_t)$ is observed.
>
> Q4: The experiment still showed that using the decomposed reward produced from OPEHF, FQE-OPEHF can have a lower MAE than just using FQE alone.
>
> R4: Thank you for this thoughtful question that leads to additional findings. Existing works have found that performance of temporal difference (TD) methods (e.g., FQE) may suffer from vanishing information [3], as well as exponential error amplification in the horizon [4]. In contrast, the immediate human rewards reconstructed by the RILR technique (section 2.3) can help resolve this problem, as it leverages latent representations learned by VLM-H to help populate reward information through the regularization (eq.(3)). Such a finding also aligns with [5], showing that by introducing a representation learning component, it can improve FQE's performance.
>
>
> Q5: The objective is on the sum of discounted rewards being the same, but there is no constraint on reward from each timestep.
>
> R5: Thank you for this thoughtful question. Table 1 of the PDF uploaded for rebuttal we report the 95% confidence interval (CI) of the reconstructed IHRs, which lie within a relatively compact range. We believe the regularization term in eq.(3) is helpful for this -- it constrained the state-action pairs that have similar latent encodings to be assigned with similar (reconstructed) IHRs, which implicitly regularize the overall scale of the reconstruction outputs and help ensure that the extreme cases would not be likely to appear. In table 1 we also listed the min and max reconstructed IHR over all trajectories per patient, and there do exist some outliers that stay further away from the 95% CI. The reason could be that either there exist (rare) state-action pairs that are critical for resulting in relatively higher/lower returns, or they are the outlier while sampling $\hat r_t^\mathcal{H}$ 's from the learned distribution $f_\theta$ (which we use Gaussian to represent). Figure 2 in the PDF further visualizes the distribution of reconstructed IHRs over the consolidated state and action space.
>
>
> [1] Namkoong et al. "Off-policy policy evaluation for sequential decisions under unobserved confounding." NeurIPS'20.
>
> [2] Chesnaye et al. "An introduction to inverse probability of treatment weighting in observational research." Clinical Kidney Journal 2022.
>
> [3] Arjona-Medina et al. "Rudder: Return decomposition for delayed rewards." NeurIPS'19.
>
> [4] Wang et al. "What are the Statistical Limits of Offline RL with Linear Function Approximation?." ICLR'20.
>
> [5] Chang et al. "Learning bellman complete representations for offline policy evaluation." ICML'22.

---

> > ### Comment · Reviewer_KHGy · 2023-08-21
> >
> > Thank you for the rebuttal -- the authors have discussed the points I've raised, and I'm satisfied by the response. I will keep my score as is.

---

> ### Author Response · Authors · 2023-08-16
> **Mid-point check-in**
>
> As we are stepping into the 2nd half of the discussion period, should the reviewer have any follow-ups, we will try out best to address them in time. If satisfied, we would greatly appreciate the reviewer to update the reviews/acknowledge our responses. We sincerely thank the reviewer again for the efforts devoted to the review process, allowing the work to be thoroughly evaluated and discussed.

---

### Official Review · Reviewer_ochq · 2023-07-27

**Soundness:** 3 good
**Presentation:** 3 good
**Contribution:** 3 good
**Rating:** 7
**Confidence:** 2

**Summary:**

The paper introduces a framework for Off-Policy Evaluation for Human Feedback (OPEHF). Off-policy evaluation allows the evaluation of learned policies on offline recorded data, which is especially valuable in scenarios where online deployment to evaluate policies is dangerous or expensive. Especially in scenarios such as healthcare, human feedback is very valuable and apparently often not very correlated with environmental rewards, making the combination of these two aspects very interesting. The paper considers the challenging case where human feedback is only available at the end of an episode without any per-step immediate human reward (IHR). To utilize existing OPE approaches, the per episode human reward is first reconstructed to a sequence of IHR by optimizing the sequence such that the cumulative discounted sum equals the human rewards, while additionally regularizing the reconstructed IHRs to align well with a latent representation of a state-action pair sequence. The resulting approach is evaluated on 2 real world scenarios, showing interesting results.


**Strengths:**

- The paper tackles a problem with realistic applications and proposes an interesting approach to do so.
- It's well written with an extensive supplementary. Even though this is sadly not my area of expertise, I felt the paper was made to be understandable.
- Results look favorable compared to simpler baselines and the method does not rely on environmental rewards being correlated with the human feedback, for which the paper gives multiple interesting example use cases.
- Code is provided.



**Weaknesses:**

- While I very much like the actual longer term studies with real humans, overall the empirical evidence still is not extremely convincing. I guess this is an inherent problem with human feedback being expensive to collect. Nevertheless, some additional experiments would of course be very valuable.


**Questions:**

- In Proposition 1 you assume that the expected value of the rewards is positive. This is obvious in the considered scenarios, but would this approach generalize to other scenarios where human feedback could indeed be negative?
- To what extend does the VLM-H hyperparameter tuning play a role here? How robust is the overall approach to less optimal setups here?
- If you could discuss further evaluation venues for the proposed method, that would be interesting, but honestly I quite like the overall paper and find it to be an interesting read. If no further bigger issues are brought up by the other reviewers, I am willing to up my rating, I would just like to keep it a little conservative given this is very much outside of my expertise.

**Limitations:**

Some limitations are discussed and no other obvious limitations or major negative societal impacts come to mind.

---

> ### Author Rebuttal · Authors · 2023-08-09
>
> We greatly appreciate the reviewer's thoughtful questions and constructive feedback. Please find our responses below. We hope they addressed the reviewer's outstanding concerns sufficiently, and we would be more than happy to further respond to any follow-ups the reviewer may have.
>
> Q1: While I very much like the actual longer term studies with real humans, overall the empirical evidence still is not extremely convincing. I guess this is an inherent problem with human feedback being expensive to collect. Nevertheless, some additional experiments would of course be very valuable
>
> R1: We greatly appreciate the reviewer's understanding toward the complexity of conducting experiments with human participants, as well as collecting feedback from them, over years of effort. We tried to focus on real-human experiments as currently there lacks a standard testing bench toward OPEHF's objective. However, we had found that recently some natural language Q&A machines [1] employ a similar evaluation fashion as our problem setup -- policies are trained over discrete environmental rewards, but are evaluated by a separate referee model that produces dense end-of-episode returns. This experiment is briefly mentioned in lines 223-225 and details can be found in Appendix E. Only moderate-correlations are found between the two types of rewards as well, and results show that our method still rigorously outperforms the baselines.
>
> Q2: In Proposition 1 you assume that the expected value of the rewards is positive. This is obvious in the considered scenarios, but would this approach generalize to other scenarios where human feedback could indeed be negative?
>
> R2: Thank you for this thoughtful question that points to one of the future directions of our work. The assumption is mainly used in the proof (Appendix A, equations (20) and (21)) that the expectation of rewards appears on both sides of the inequality. With some initial exploration, we might be able to relax this assumption in the future by using a different theoretical framework for the proof, but with the price of obtaining an asymptotic bound, instead of the strict variance reduction property as shown in proposition 1. On the other hand, if the lower and upper limit of rewards are given *a priori*, one may apply transformations over the rewards so that they satisfy the assumption. In the meantime, please note that several downstream OPE methods may intrinsically require the rewards to follow specific properties. For example, some FQE methods [2, 3] assume that rewards are in $[0,1]\subset\mathbb{R}$, and a doubly-robust (DR) method [4] assumes that the lower and upper limit of the rewards are known.
>
> Q3: To what extent does the VLM-H hyperparameter tuning play a role here? How robust is the overall approach to less optimal setups here?
>
> R3: Good question. We have done further analyses to this end -- please see Figure 1 in the PDF uploaded for rebuttal. The **VLM-H is mainly used to facilitate the regularization** (eq.(3)), where the state-action pairs that lead to similar embeddings should get similar (reconstructed) immediate human rewards (IHRs). So we find that the overall performance is not significantly bounded by VLM-H. Specifically, in Figure 1 we visualize the latent space of multiple sub-optimal VLM-H's trained with different hyper-parameter sets. The latent space in general demonstrates decent clustering behavior, if the hyper-parameters are not too off, which can be guaranteed by inspecting the visualization of a holdout validation set. Sub-optimality in VLM-H would only lead to less than 5% fluctuation in performance in general, which would be considered insignificant as our method outperforms baselines generally with great margins.
>
> Q4: If you could discuss further evaluation venues for the proposed method, that would be interesting, but honestly I quite like the overall paper and find it to be an interesting read. If no further bigger issues are brought up by the other reviewers, I am willing to up my rating.
>
> R4: What makes the evaluation of OPEHF challenging, would be the fact that the goal is to estimate participants' feedback toward their own experience (e.g., patient's satisfaction after treatment). Unlike frameworks like RLHF for language model training [5], where one can hire 3rd-party raters to provide feedback simply by following a predefined rubric (without the need of domain expertise); in human-centric domains like healthcare or education, professionals that are thoroughly trained (e.g., doctors, professors, instructors) over years may only be eligible to rate, other than the participants themselves. As a result, we currently also invest significant effort toward building a platform that provides de-identified and privacy-protected dataset (e.g., the one we used in this paper) through API access, as well as allow verified professionals to provide feedback/ratings. Moreover, once the platform scales, we can train a referee model (similar to the one used in the automated Q&A machine), to capture how professionals would examine new trajectories and provide ratings directly. Hopefully, in the near future, researchers can leverage the platform to evaluate OPEHF-related works in an standardized manner.
>
> [1] Snell et al. "Offline RL for Natural Language Generation with Implicit Language Q Learning." ICLR'22.
>
> [2] Zhang et al. "Off-policy fitted q-evaluation with differentiable function approximators: Z-estimation and inference theory." ICML'22.
>
> [3] Hao et al. "Bootstrapping fitted q-evaluation for off-policy inference." ICML'21.
>
> [4] Thomas et al. "Data-efficient off-policy policy evaluation for reinforcement learning." ICML'16.
>
> [5] Ziegler et al. "Fine-tuning language models from human preferences." arXiv:1909.08593 (2019).

---

> ### Author Response · Authors · 2023-08-16
> **Mid-point check-in**
>
> As we are stepping into the 2nd half of the discussion period, should the reviewer have any follow-ups, we will try out best to address them in time. If satisfied, we would greatly appreciate the reviewer to update the reviews/acknowledge our responses. We sincerely thank the reviewer again for the efforts devoted to the review process, allowing the work to be thoroughly evaluated and discussed.

---

> > ### Comment · Reviewer_ochq · 2023-08-16
> > **Post rebuttal opinion**
> >
> > Thank you for answering all the questions and concerns. I was mainly curious to see if reviewer 1oyy would still reply. Even though this is not my area of expertise, I think you addressed the concerns thoroughly. Unless reviewer 1oyy is willing to spearhead the rejection of this paper, I am recommending this paper to be accepted and I upped my rating accordingly.

---

> > > ### Author Response · Authors · 2023-08-16
> > > **Thank you**
> > >
> > > We sincerely thank the reviewer for the prompt response, as well as the patience and efforts dedicated to evaluating our work thoroughly.

---

### Author Rebuttal · Authors · 2023-08-09

# General Response

We sincerely thank the reviewers for the time and efforts dedicated to the review process.

We feel that currently the main discrepancy centered around the evaluation of OPEHF. While reviewers KHGy and cuXN pointed that our real-world experiments are convincing and comprehensive, reviewer 1oyy seemed to prefer more evaluation studies. Reviewer ochq stayed neutral while conceded that obtaining data with participant's feedback would be intrinsically challenging.

We agree that currently there may lack a standard testing bench for OPEHF across the community, as (to the best of our knowledge) we are the first that introduces the framework that revives existing OPE methods toward estimating the human feedback/returns that are only provided at the end of each episode (that are not correlate well with environmental rewards); our method addresses a long-standing challenge in the community of OPE (as pointed by KHGy, ochq and cuXN). However, please note that our two real-world experiments are backed with substantial data collected from years of follow-ups, with results showing our method significantly outperforms over baselines. **We did also supplement these experiments with a simulation study on natural language Q&A machine** in Appendix E, which employs similar schema that policies are trained on discrete environmental rewards, but evaluated against dense returns issued by a separate referee model. Our method significantly outperform the baselines in this setting as well.

We had also discussed, in section 4, that OPEHF is mainly evaluated over *direct preference* provided by participants themselves, which is different than the *ranked preference* used in RLHF, hence made it intractable to evaluate over RLHF's setup -- the participants cannot revisit the same procedure multiple times (patients may not undergo the same surgeries several times and rank the experiences). However, on the other hand, we discussed in the response to reviewer ochq's Q4 in regards to facilitating future venues for evaluating OPEHF-related works, where we are currently building a platform that allows verified 3rd party professionals (doctors and professors) to provide direct feedback to logged trajectories, to further consolidate the standardized evaluation process for future works along this line.

Again, we greatly appreciate the reviewers' and chairs' efforts toward the review and rebuttal processes, allowing the work to be thoroughly discussed and evaluated.

------------------------------------------------------------------------------------------------------------------------------------

At last, please find in attached the PDF that contains figures and tables, corresponding to the additional empirical analyses used to address some of the individual comments below.

---

### Decision · Program_Chairs · 2023-09-21

**Decision:**

Accept (poster)

**Comment:**

This paper studies the problem of offline policy evaluation with human feedback. Under this setting, human feedback as a reward is given only at the end of an episode. It results in a sparse reward environment whose data is difficult to collect in practice such as in healthcare. This paper proposes to first learn a surrogate per-step immediate human reward (IHR) that decomposes the final human feedback, and then apply existing OPE methods on the learned reward.

This paper makes a good progress on this challenging and realistic problem. Multiple reviewers consider its two real-world experiments a strength. Some reviewer is worried that the dataset is too small for thorough evaluation but the author explains it's due to the lack of data in real problems that takes years to collect.

The main concern comes from the fact that the main method can be considered as a type of reward shaping. It is not compared to prior works along this perspective, such as RUDDER. While authors argue RUDDER was devised for policy optimization rather than OPE and is not applicable, the reviewer consider the reward decomposition idea of RUDDER still relevant and its variants should be detailed discussed in this work.

The reviewers also made multiple insightful comments including the design choice of the proposed bi-directional LSTM, the MDP formulation and Markovian assumption, the limitation on handling conflicting human feedbacks. While these comments does not undermine the contribution of this paper to tackling the OPE-HF problem, the authors should take them seriously. Incorporate those comments in the revision and providing a clear discussion on the limitations would definitely improve the work.